# DIFFERENTIABLE LOGIC PROGRAMMING FOR PROBABILISTIC REASONING

## ABSTRACT

This paper studies inductive logic programming for probabilistic reasoning. The key problems, i.e. learning rule structures and learning rule weights, have been extensively studied with traditional discrete searching methods as well as recent neural-based approaches. In this paper, we present a new approach called Differentiable Logic Programming (DLP), which provides a flexible framework for learning first-order logical rules for reasoning. We propose a continuous version of optimization problem for learning high-quality rules as a proxy and generalize rule learning and forward chaining algorithms in a differentiable manner, which enables us to efficiently learn rule structures and weights via gradient-based methods. Theoretical analysis and empirical results show effectiveness of our approach.

## 1 INTRODUCTION

Learning to reason and predict is a fundamental problem in the fields of machine learning. Representative efforts on this task include neural networks (NN) and inductive logic programming (ILP). The NNs and ILP methods represent learning strategies of two extremes: the ideas behind NNs are to use fully differentiable real-valued parameters to perceive the patterns of data, while in the fields of ILP, we search for determined and discrete structures to match the patterns of data. Over the years, the former approaches, i.e. neural-based methods, have achieved state-of-the-art performance in solving tasks from many different fields, while the latter ones have fallen behind due to their inherited inferior in fitting noisy and probabilistic data.

However, it was pointed out that there is a debate over the problems of systematicity and explanability in connectionist models, as they are black-box models that are hard to be explained. To tackle the problem, numerous methods have been proposed to combine the advantages of both connectionist and symbolic systems. Most existing efforts focus on two different manners: using logic to enhance neural networks and using neural networks to help logical reasoning. The former approaches (Rocktäschel & Riedel (2017), Minervini et al. (2020c), Vedantam et al. (2019), Dong et al. (2019)) modify the structures of NNs to capture some features of logic. Some of them, known as neural theorem provers (Rocktäschel & Riedel (2017), Minervini et al. (2020a)), represent entities with embedding vectors that implies the semantics of them. Further more, they absorb symbolic logical structures into the neural reasoning framework to enhance the expressiveness of the models. For example, to prove the existence of $(grandfather, Q, Bart)$ where $Q$ is the target entity we wish to find, these systems use logic rules such as $grandfather \leftarrow father\_of, parent\_of$ to translate the original goal $(grandfather, Q, Bart)$ into subgoals that can be subsequently proved by operating on entity embeddings. Thus, the expressiveness and interpretability of the systems is improved with the help of logic.

The latter approaches (Yang et al. (2017), Xiong et al. (2017) Sadeghian et al. (2019), Qu et al. (2021)) enhance traditional inductive logic programming with the help of neural networks. Generally, they use different techniques to solve the key problems of ILP, which is to learn structures of logical rules from exponential large space. Some of them (Yang et al. (2017), Sadeghian et al. (2019), Yang & Song (2020)) approximate the evaluation of all possible chain-like logic rules in a single model, making learning of the model differentiable. However, as mentioned in Sadeghian et al. (2019), these models inevitable give incorrect rules with high confidence values due to the low-rank approximation of evaluating exponential many logic rules at the same time, which also

makes it hard to identify high-quality logic rules and explain the predictions made by these models. The other line of the research (Yang & Song (2020), Qu et al. (2021)) propose different methods to generate high-value rules such as reinforce learning and EM algorithms. However, since structure learning of logical rules is a very hard problem, they are limited in only searching chain-like horn clauses, which is less expressive and general.

In this paper, we propose a novel differentiable programming framework, called Differentiable Logic Programming (DLP), to build a bridge between the ideas of differentiable programming and symbolic reasoning. Our approach enjoys the merits of connectionist systems, i.e., high expressiveness and easy to learn, as well as the merits of ILP systems, i.e., explanability and clear structures for decision making. We study the construction of a probabilistic reasoning model, and discuss the properties of valuable logic rules. Based on that, we propose a novel rule learning framework that approximates the combinatory search problem with a continuous relaxation which enables us to learn structures of logic rules via a differentiable program. Once valuable rules are learnt, we can further fine-tune the rule weights and perform probabilistic forward chaining to predict the existence of unobserved terms.

## 2 RELATED WORK

Our work is related to previous efforts on Inductive Logic Programming (ILP) fields and their extensions. Representative methods of ILP includes FOIL (Quinlan (2004)), MDIE (Muggleton (2009)), AMIE (Galárraga et al. (2015)), Inspire (Schüller & Kazmi (2018)), RLvLR (Omran et al. (2018)) and so on. Generally, these methods search for logic rules in exponential large space to obtain valuable logic rules and make predictions based on them. However, despite the well-designed searching algorithms and pruning techniques, these methods suffer from their inherent limitations of relying on discrete counting and predefined confidence.

More recently, different learning algorithms have been proposed to overcome the drawbacks of ordinary ILP methods. Many of them consider a special kind of ILP tasks namely knowledge graph completion, where most of the proposed methods (Yang et al. (2017), Rocktäschel & Riedel (2017), Sadeghian et al. (2019), Minervini et al. (2020b), Yang & Song (2020), Qu et al. (2021)) focus on learning chain-like rules, and these methods use different learning strategies to learn valuable rules. Some of them are based on reinforcement learning (Xiong et al. (2017), Chen et al. (2018), Das et al. (2018), Lin et al. (2018), Shen et al. (2018)), and they train agents to find the right reasoning paths to answer the questions in knowledge graphs. Qu et al. (2021) uses recurrent neural networks as rule generators and train them with EM algorithms. Yang et al. (2017), Sadeghian et al. (2019) and Yang & Song (2020) propose end-to-end differentiable methods, which can be trained efficiently with gradient-based optimizers. These methods are similar in spirit with our approach, as they claim to be able to learn rule structures in a differentiable manner. However, what they actually do is to find a low-rank tensor approximation for simultaneous execution of all possible rules of exponential space with different confidence scores, and by doing so they suffer from the risk of assigning wrong rules with high scores (Sadeghian et al. (2019)). Also, although Yang & Song (2020) claims that their attentions usually becomes highly concentrate after convergence, there is no theoretical guarantee so extracting logic rules implying these model could be problematic because there might be exponential potential rules that have confidence scores higher than zero. The parameters learnt by these models are dense vectors thus they suffer from the problem of explainability. Compared with them, our method is able to generate sparse solutions that explicitly learns logic rules for reasoning with a more flexible rule search space while keeping the rule learning procedure differentiable.

There are other methods that focus on different types of ILP problems. Lu et al. (2022) treats relation prediction task as a decision making process, and they use reinforcement learning agents to select the right paths between heads and tails. Our approach is more general and is able to deal with different tasks. Rocktäschel & Riedel (2017) and Minervini et al. (2020b) propose a generalized version of backward chaining with the help of neural embedding methods, and show great performance on both relation prediction and knowledge graph completion tasks. Compared to them, our approach doesn't require the help of embeddings, thus our predictions are more explainable.

There are also interesting methods based on embedding and neural networks (Bordes et al. (2013), Wang et al. (2014), Yang et al. (2015), Nickel et al. (2016), Trouillon et al. (2016), Cai & Wang

(2018), Dettmers et al. (2018), Balazevic et al. (2019), Sun et al. (2019), Teru et al. (2020), Zhu et al. (2021)). Since they are less relevant to logic reasoning, we do not cover them in details here.

## 3 PRELIMINARY

### 3.1 FIRST-ORDER LOGIC

This paper focuses on learning first-order logic (FOL) rules for reasoning. Across this paper, we assume predicates are from a countable universe $\mathcal{P}$ where we use uppercase $P, Q, ... \in \mathcal{P}$ to represent predicates. We use $a, b, c, ... \in \mathcal{V}$ to represent constants and $x, y, z$ to represent variables. An example of grammars of FOL applied in the experiments of this paper are:

$$\varphi(x) := P(x) \mid \varphi(x, x) \mid \exists y : \varphi(y) \wedge \varphi(x, y),$$
$$\varphi(x, y) := P(x, y) \mid \varphi(x) \wedge \varphi(y) \wedge \varphi(x, y) \mid \exists z : \varphi(x, z) \wedge \varphi(z) \wedge \varphi(z, y). \tag{1}$$

Grammars are critical for ILP systems, because they not only define the syntax of FOL formulas, but also determine the expressive power and search space of FOL formulas. However, in this paper we will not restrict the specific formulation of the grammar. Instead, we use a common formulation to represent them and our approach is equivalently applicable to any reasonable grammar:

$$\varphi(\mathbf{x}) := P(\mathbf{x}) \mid F_1(\mathbf{x}) \mid F_2(\mathbf{x}) \mid F_3(\mathbf{x}) \mid ... \tag{2}$$

Also, Eq. 1 is frequently used to demonstrate the ideas in the following sections. We use $\mathbf{x}$ to represent a tuple of variables, $\mathbf{v}$ for a tuple of constants for notation simplification given it's clear in the context. $F_i$ represents a possible format that $\varphi$ could take. For example, in the definition of $\varphi(x)$ in Eq. 1, we have $\mathbf{x} = x$ and $F_1(\mathbf{x}) := \varphi(x, x)$, $F_2(\mathbf{x}) := \exists y : \varphi(y) \wedge \varphi(x, y)$.

**Logic classifiers and logic rules** Logic formulas $\varphi$ can be regarded as classifiers (Barceló et al. (2020)). For example, consider $\varphi(x) := \exists y : \text{Red}(y) \wedge \text{Edge}(x, y)$ and $\text{Blue}(x) \leftarrow \varphi(x)$. $\varphi$ can be regarded as a classifier, where we have $\varphi(x) = 1$ for nodes $x$ with red neighbors and 0 otherwise. Generally, logic classifiers take (set of) entities (e.g. node $x$) as input and compute the output (e.g. $\varphi(x)$) by grounding the logic formula on the background statements (e.g. $\{\text{Edge}(x, y), \text{Red}(y)\}$). $\text{Blue}(x) \leftarrow \varphi(x)$ is a logic rule that tells us the rule head $\text{Blue}(x)$ can be concluded if the rule body $\varphi(x)$ is satisfied. In the fields of probabilistic reasoning, each logic rule can be assigned with a weight indicating the degree of certainty of the rule.

**Forward chaining and backward chaining** Forward chaining methods are critical in automated deduction, as they enable us to repeatedly deduce new lemmas from known theorems. Forward chaining starts from known conditions and logic rules, and move forward towards a conclusion by applying the logic rules. Then, they absorb the deduced conclusions into known conditions and apply the rules again until there are no further new conclusions deduced. Backward chaining methods are the opposite of forward chaining, as they move backwards from the conclusions to the potential conditions implied by the rules.

### 3.2 PROBLEM STATEMENT

This paper studies probabilistic inductive logic programming. The input data is a tuple $(\mathcal{S_B}, \mathcal{S_P}, \mathcal{S_N})$ where $\mathcal{S_B}, \mathcal{S_P}, \mathcal{S_N}$ are sets of ground atoms of the form $\{P_1(\mathbf{v}_1), P_2(\mathbf{v}_2), ...\}$, $\mathcal{S_B}$ is a set of background assumptions, $\mathcal{S_P}$ is a set of positive instances, and $\mathcal{S_N}$ is a set of negative instances. The target is to construct a model so that when applied to $\mathcal{S_B}$, it produces the positive conclusions in $\mathcal{S_P}$, as well as rejecting the negative instances in $\mathcal{S_N}$. This naturally leads to the following problems:

- **Rule Mining.** Our model is based on logic rules, and one key problem is finding useful rules that produce the results in $\mathcal{S_P}$ when grounded on $\mathcal{S_B}$.

- **Probabilistic Reasoning.** It is often infeasible to directly perform forward chaining with logic rules when input data is noisy. A rule-based inference model $p(Q(\mathbf{x}) \in \mathcal{S_P} | \mathcal{S_B})$ is needed to take the uncertainty of logic rules into account.

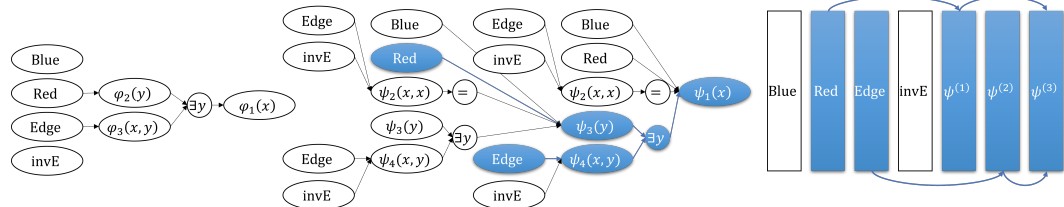

Figure 1: Illustration of logic classifiers (left) and DLP framework (middle, right). The target rule is $\text{Blue}(x) \leftarrow \exists y : \text{Red}(y) \wedge \text{Edge}(x, y)$. In the left figure, $\varphi_1(x)$ directly captures the target rule. In the middle figure, the model is constructed by extending the unary node $\psi_1(x)$ with tree structure. In the right figure, the model is constructed by stacking multiple layers. (see Sec. 4.2 for detailed description). Both these structures are constructed with the grammar in Eq. 1. $\psi_1$ learns to identify the target logic rule via gradient descent and converges at the correct (colored as blue) reasoning paths.

# 4 MODEL

In this section, we introduce our proposed Differentiable Logic Programming (DLP) framework. The general idea is to use differentiable programs to solve the problems of rule mining and learning the prediction model.

As mentioned in Sec. 3, the learning problems require us to identify important logic classifiers from discrete space and assign feasible weights to them. In this paper, we introduce a differentiable module called Logic Perceptron (LP) to help us deal with the problem. Given a grammar of logic classifiers $\varphi$, we provide a method to construct corresponding LPs $\psi$ that is able to capture any $\varphi$ with limited size. The LPs are stacked as a network to capture more complex logic classifiers as well as being end-to-end differentiable. We further propose a new optimization problem whose solutions are sparse so that each local optimal of it corresponds to the symbolic structure of a logic rule being revealed. Moreover, these learnt LPs are organized into a prediction model that generalizes forward chaining and can be learnt by maximizing the likelihood. Both these optimization problems are continuous and differentiable w.r.t. parameters, which makes them can be solved via gradient-based methods. Figure 1 presents a brief illustration of general ideas behind our model. Next, we introduce the details of our approach.

## 4.1 OVERVIEW

In this section we first introduce the general ideas behind our approach, as well as highlighting the key challenges in our learning framework. The details of model implementation are discussed in the next sections. We start with an example which illustrates the general ideas of probabilistic reasoning as well as providing the intuitions and motivations behind our approach.

**Example 4.1** (Human Reasoning). *Consider the query* "*Are $a$ and $b$ friends?*". *In this case, we have $Q(\mathbf{x}) = \text{Friend}(a, b)$. To answer the query, one may first ask* "*Do $a$ and $b$ know each other?*" *which corresponds to a logical classifier $\varphi_1(a, b) = \text{Know}(a, b)$. If $\varphi_1(a, b) = 1$, our confidence in $x$ and $y$ being friends is increased, and we call this $\varphi_1$ proved $Q$, also $Q$ is the target predicate of $\varphi_1$, denoted as $Q \leftarrow \varphi_1$. With the answer of $\varphi_1$, one may continue to ask $\varphi_2$:* "*Do $a$ and $b$ live in the same town?*", *..., where each evaluation of $\varphi_i$ serves as an evidence that proves the existence of $Q$.*

In the example, we use different logic classifiers to prove the query from different aspects, and all the classifiers used are highly relevant to the target predicate $\text{Friend}$. This makes sense because our belief in $x$ and $y$ being friends is higher when we realize they know each other, which corresponds to a relative high value of $p(\text{Friend}(x, y) | \text{Know}(x, y) = 1)$, but irrelevant facts such as "they both drink water" won't help. In fact, this simple example illustrates the overall ideas of our model, formally described as follows.

**Proposition 4.2** (Properties of $\Phi$). *Given the input data $(\mathcal{S}_\mathcal{B}, \mathcal{S}_\mathcal{P}, \mathcal{S}_\mathcal{N})$, let $Q$ be a $d$-ary target predicate, $\Phi_Q = \{\varphi_0, \varphi_1, ..., \varphi_{L-1}\}$ be a set of logic classifiers and $p_0 \in (0, 1]$ a fixed threshold. With the following statements being satisfied: (1) We start from $l = 0$ and let $\mathcal{S}_\mathcal{P}^{(0)} = \mathcal{S}_\mathcal{P}$; (2) For $\varphi_l$, its precision: $p(Q(\mathbf{x}) \in \mathcal{S}_\mathcal{P}^{(l)}|\varphi_l(\mathbf{x}) = 1) = \frac{\sum_{\mathbf{x} \in \mathcal{V}^d} \varphi_l(\mathbf{x}) \, \mathbf{1}_{Q(\mathbf{x}) \in \mathcal{S}_\mathcal{P}^{(l)}}}{\sum_{\mathbf{x} \in \mathcal{V}^d} \varphi_l(\mathbf{x}) \, \mathbf{1}_{Q(\mathbf{x}) \in \mathcal{S}_P^{(l)} \cup \mathcal{S}_N}} \geq p_0$; (3) We let $\mathcal{S}_\mathcal{P}^{(l+1)} = \mathcal{S}_\mathcal{P}^{(l)} \setminus \{Q(\mathbf{x}) \mid \mathbf{x} \in \mathcal{V}^d, \varphi_l(\mathbf{x}) = 1\}$ and increase $l$ by 1 and go to (2) again until $l = L$; (4) $\mathcal{S}_\mathcal{P}^{(L)} = \emptyset$. Then, there exists a prediction model $p(Q(\mathbf{x}) \in \mathcal{S}_\mathcal{P}|\mathcal{S}_\mathcal{B}) = f(\varphi_0(\mathbf{x}), \varphi_1(\mathbf{x}), ..., \varphi_{L-1}(\mathbf{x}))$ such that its error rate satisfies:*

$$\mathrm{Err}\left[f; (\mathcal{S}_\mathcal{B}, \mathcal{S}_\mathcal{P}, \mathcal{S}_\mathcal{N})\right] \leq \frac{(1 - p_0)N_Q}{p_0 N} \leq 1 - p_0, \tag{3}$$

*where $N_Q = \sum_{\mathbf{x} \in \mathcal{V}^d} \mathbf{1}_{Q(\mathbf{x}) \in \mathcal{S}_\mathcal{P}}$ and $N = \sum_{\mathbf{x} \in \mathcal{V}^d} \mathbf{1}_{Q(\mathbf{x}) \in \mathcal{S}_\mathcal{P} \cup \mathcal{S}_\mathcal{N}}$.*

Prop. 4.2 implies how we learn such logic classifiers for proving $Q$. Generally our rule learning procedure can be seen as a variant of boosting method where each rule is regarded as a weak classifier, but we focus on the rule precision rather than misclassification rate. This is because (1) in most situations the input data is very sparse that one can obtain a small misclassification rate by simply always predicting false, and (2) often a single logic rule is only able to prove a relative small portion of the positive instances and a large number of logic rules are often needed to make complete predictions of the target predicates. Thus, our learning procedure is formally described as follows.

**Rule Learning Framework**   Given the input data $(\mathcal{S}_\mathcal{B}, \mathcal{S}_\mathcal{P}, \mathcal{S}_\mathcal{N})$, suppose we are to learn $L$ rules for each target predicate $Q \in \mathcal{P}$. We first start with an empty rule set $\Phi_Q = \emptyset$ for each $Q$, and assign each instance $Q(\mathbf{v})$ in $\mathcal{S}_\mathcal{P}$ and $\mathcal{S}_\mathcal{N}$ with a weight $w_{Q(\mathbf{v})} = 1$ (This corresponds to the statement (1) in Prop 4.2). Then, we perform the follow steps recursively: we first find a logic classifier $\varphi$ having high precision on the weighted data, i.e.,

$$\max_\varphi \quad p(Q(\mathbf{x}) \in \mathcal{S}_\mathcal{P}|\varphi(\mathbf{x}) = 1) = \frac{\sum_{\mathbf{x} \in \mathcal{V}^d} \mathbf{1}_{Q(\mathbf{x}) \in \mathcal{S}_P} \, w_{Q(\mathbf{x})} \, \varphi(\mathbf{x})}{\sum_{\mathbf{x} \in \mathcal{V}^d} \mathbf{1}_{Q(\mathbf{x}) \in \mathcal{S}_P \cup \mathcal{S}_N} \, w_{Q(\mathbf{x})} \, \varphi(\mathbf{x})}. \tag{4}$$

This generalizes the statement (2) in Prop. 4.2. We add $\varphi$ into $\Phi_Q$. Then, we evaluate $\varphi$ on $\mathcal{S}_\mathcal{B}$, and for instances $\mathbf{v}$ where $\varphi(\mathbf{v}) = 1$, if $Q(\mathbf{v}) \in \mathcal{S}_\mathcal{P}$, we let $w_{Q(\mathbf{v})} \leftarrow \tau_1 \, w_{Q(\mathbf{x})}$; if $Q(\mathbf{v}) \in \mathcal{S}_\mathcal{N}$, we let $w_{Q(\mathbf{v})} \leftarrow \tau_2 \, w_{Q(\mathbf{v})}$, where $\tau_1, \tau_2 \in [0, +\infty)$ are fixed values. Note that this step generalizes statement (3) in Prop. 4.2, and statement (3) is a special case of the above step where we let $\tau_1 = 0, \tau_{2=}1$. This procedure is then repeated for $L$ times.

Now we have introduced the overall learning framework of our approach except two problems: how we identify high-precision logic classifiers (Eq. 4) and how we learn the prediction model $p(Q(\mathbf{x}) \in \mathcal{S}_\mathcal{P}|\mathcal{S}_\mathcal{B})$. In this paper, we propose a differentiable model to solve these problems. In the next sections we discuss the implementation of our model in details. In Sec. 4.2, we introduce logic perceptrons (LP) as building blocks of our model as well as how we stack them together to express more complex logic classifiers. In Sec. 4.3 we introduce how use LPs to perform probabilistic reasoning. Then, we present our methods for learning the model in Sec. 4.4.

## 4.2   Logic Perceptrons

A Logic Perceptron (LP) is a differentiable model that generalizes the ordinary logic classifiers into continuous space. Given the grammar of logic classifiers:

$$\varphi(\mathbf{x}) := A(\mathbf{x}) \mid \mathrm{F}_1(\mathbf{x}) \mid \mathrm{F}_2(\mathbf{x}) \mid \mathrm{F}_3(\mathbf{x}) \mid ... \mid \mathrm{F}_\mathrm{K}(\mathbf{x}), \tag{5}$$

We provide two methods for constructing corresponding LPs given the grammar 5, which are tree-structured LPs (LP-tree) and layer-structured LPs (LP-layer). The main difference is that LP-tree strictly satisfies the constraints discussed in Sec. 4.4 while LP-layer is more compressed. As will be shown in Sec. 5, they produce similar results in the experiments.

**LP-tree**  We first define the correspondence between LPs $\psi$ and logic classifiers $\varphi$ as follows.

$$\psi(\mathbf{x}; \alpha) = [\mathcal{F}_1(\mathbf{x}; \alpha), \mathcal{F}_2(\mathbf{x}; \alpha), \mathcal{F}_3(\mathbf{x}; \alpha), ... \mathcal{F}_K(\mathbf{x}; \alpha)] \ \boldsymbol{w}^{\alpha},$$

$$s.t. \quad \sum_{i=1}^{K} w_i = 1, \quad w_i \geq 0 \ for \ i = 1, 2, ... K \ , \tag{6}$$

where $\boldsymbol{w} \in \mathbb{R}^K$ is an attention vector, $\alpha \in \mathbb{R}$ is a hyperparameter that helps to keep the sparsity of the model. The functionalities of $\alpha$ are discussed in Sec. 4.4, and often in experiments we set $\alpha = 1$. $\boldsymbol{w}^{\alpha}$ is an element-wise exponent applied to $\boldsymbol{w}$. The evaluation of each $\mathcal{F}_i(\mathbf{x}; \alpha)$ in Eq. 6 is corresponded to each $\mathrm{F}_i(\mathbf{x})$ in Eq. 5, where we define

$$
\begin{aligned}
\mathrm{F}_i(\mathbf{x}) &:= \mathrm{F}_j(\mathbf{x}) \wedge \mathrm{F}_k(\mathbf{x}) &&\iff \mathcal{F}_i(\mathbf{x}; \alpha) = \mathcal{F}_j(\mathbf{x}; \alpha)\mathcal{F}_k(\mathbf{x}; \alpha), \\
\mathrm{F}_i(\mathbf{x}) &:= \mathrm{F}_j(\mathbf{x}) \vee \mathrm{F}_k(\mathbf{x}) &&\iff \mathcal{F}_i(\mathbf{x}; \alpha) = [\mathcal{F}_j(\mathbf{x}; \alpha), \mathcal{F}_k(\mathbf{x}; \alpha)] \left(\boldsymbol{w}^{(i)}\right)^{\alpha}, \\
\mathrm{F}_i(\mathbf{x}) &:= \neg\mathrm{F}_j(\mathbf{x}) &&\iff \mathcal{F}_i(\mathbf{x}; \alpha) = 1 - \mathcal{F}_j(\mathbf{x}; \alpha), \\
\mathrm{F}_i(\mathbf{x}) &:= \exists\mathbf{y} : \mathrm{F}_j(\mathbf{x}, \mathbf{y}) &&\iff \mathcal{F}_i(\mathbf{x}; \alpha) = \sum_{\mathbf{y}} \mathcal{F}_j(\mathbf{x}, \mathbf{y}; \alpha), \\
\mathrm{F}_i(\mathbf{x}, \mathbf{y}) &:= \mathrm{F}_j(\mathbf{x}) &&\iff \mathcal{F}_i(\mathbf{x}, \mathbf{y}; \alpha) = \mathcal{F}_j(\mathbf{x}; \alpha), \\
\mathrm{F}_i(\mathbf{x}) &:= \varphi(\mathbf{x}) &&\iff \mathcal{F}_i(\mathbf{x}; \alpha) = [P_1(\mathbf{x}), P_2(\mathbf{x}), ..., P_{|\mathcal{P}|}, \tilde{\psi}(\mathbf{x})] \left(\boldsymbol{w}^{(i)}\right)^{\alpha},
\end{aligned} \tag{7}
$$

where for each $\boldsymbol{w}^{(i)}$ we have $\sum_j w_j^{(i)} = 1$ and $w_j^{(i)} \geq 0$. $\tilde{\psi}$ is a pointer to another LP. We exclude the universal quantifier $\forall$ because this can be equivalently expressed by $\neg \exists \neg$.

To construct more complex and expressive LPs, we can generate arbitrary numbers of LPs within a tree structure. Initially, we create a $\psi_0(\mathbf{x})$ as a root node. As shown in Eq. 7, the evaluation of $\psi_0$ requires its pointers $\tilde{\psi}$ to be explicitly assigned, so we create a new LP for each pointer $\tilde{\psi}$ of $\psi_0$ to be a child of $\psi_0$. The same procedure is performed on the leaf nodes of the tree for arbitrary times while expanding the depth of the tree. Once we reached the desired depth, we simply assign empty nodes as $\tilde{\psi}$ for the leaf nodes to terminate the construction procedure. These LPs compose a LP-tree where $\psi_0$ serves as the output of the tree. Figure 1 illustrates this procedure.

**LP-layer**  Generally layer structured LPs are similar with tree structured LPs except that now we stack multiple LPs linearly as layers. Suppose for LP-layer we have a total number of $L$ LPs $\{\psi^{(1)}, \psi^{(2)}, ..., \psi^{(L)}\}$. The evaluation of each LP is exactly the same as in Eq. 6 and 7 except that for each LP $\psi^{(l)}$ in LP-layer we have

$$\mathrm{F}_i(\mathbf{x}) := \varphi(\mathbf{x}) \iff \mathcal{F}_i(\mathbf{x}; \alpha) = [P_1(\mathbf{x}), P_2(\mathbf{x}), ..., \psi^{(1)}(\mathbf{x}; \alpha), ..., \psi^{(l-1)}(\mathbf{x}; \alpha)] \left(\boldsymbol{w}^{(i)}\right)^{\alpha}, \tag{8}$$

, thus $\psi^{(l)}$ is able to access the layers before it $P_1, P_2, ..., \psi^{(1)}, ..., \psi^{(l-1)}$, and the LPs are organized as layers illustrated in Fig. 1, where $\psi^{(L)}$ serves as the output of the layers. An advantage of LP-layer is that it's very simple to extend the model: we only need to stack more layers. The following proposition states the expressiveness of these two construction approaches.

**Proposition 4.3** (Expressiveness of LP-tree and LP-layer). *Given a grammar of logic classifiers of the form in Eq. 5, suppose a logic classifier $\varphi$ is constructed by recursively applying the grammar for $N > 0$ times, then we have:*
*(1) In worst cases LP-tree with $O(K^N)$ LPs can express $\varphi$;*
*(2) LP-layer with $N$ LPs can express $\varphi$.*

## 4.3 INFERENCE

We now discuss how we infer $p(Q(\mathbf{x}) \in \mathcal{S}_{\mathcal{P}}|\mathcal{S}_{\mathcal{B}})$ for every $\mathbf{x}$ by generalizing forward chaining. In this section, we assume we have learnt a set of $\Psi_Q = \{\psi_1, \psi_2, ...\}$ for every $Q \in \mathcal{P}$, and the goal is to infer the (unknown) positive / negative instances in $\mathcal{S}_{\mathcal{P}}, \mathcal{S}_{\mathcal{N}}$. For each predicate $Q \in \mathcal{P}$ and every $\mathbf{x}$, we let $Q^{(0)}(\mathbf{x}) = 1$ if $Q^{(0)}(\mathbf{x}) \in \mathcal{S}_{\mathcal{B}}$ and $Q^{(0)}(\mathbf{x}) = 0$ otherwise. Then, at iteration $t$, we:
(1) Evaluate LPs: We evaluate every $\psi \in \Psi_Q$ on $Q^{(t-1)}$ at every $\mathbf{x} \in \mathcal{V}^d$. This procedure costs

$O(|\mathcal{V}|^d R_\Psi(|\mathcal{P}| + N)N)$ where $N$ is the size of the network, $d$ is the maximum arity of predicates and LPs when evaluating $\psi$, $R_\Psi$ is the amount of rules, i.e., the total size of $\Psi_Q$ for each $Q$.
(2) Update inferences: We update our inferences about $Q(\mathbf{x})$ for each $Q \in \mathcal{P}$ and $\mathbf{x}$. This procedure costs $O(|\mathcal{V}|^d R_\Psi)$.

$$\hat{Q}(\mathbf{x}) = \mathrm{sigmoid}\left(\mathrm{Update}\left(\{\psi_Q(\mathbf{x})|\psi_Q \in \Psi_Q\}\right)\right),$$
$$Q^{(t)}(\mathbf{x}) = \max\{\hat{Q}(\mathbf{x}), Q^{(0)}(\mathbf{x})\}, \tag{9}$$

where $\mathrm{Update}$ is the function that specifies how we update the predictions based on the groundings of $\psi_Q \in \Psi_Q$. The common implementation of the update function used in this paper is

$$\mathrm{Update}\left(\{\psi_Q(\mathbf{x})|\psi_Q \in \Psi_Q\}\right) = \sum_{\psi_Q \in \Psi_Q} w_{\psi_Q} \psi_Q(\mathbf{x}), \tag{10}$$

but any differentiable update functions (MLP, etc.) is also applicable. After $T$ rounds of iterations, we directly pick the values of $Q^{(T)}(\mathbf{x})$ as an approximation of $p(Q(\mathbf{x}) \in \mathcal{S}_\mathcal{P}|\mathcal{S}_\mathcal{B})$, while $p(Q(\mathbf{x}) \in \mathcal{S}_\mathcal{N}|\mathcal{S}_\mathcal{B}) = 1 - p(Q(\mathbf{x}) \in \mathcal{S}_\mathcal{P}|\mathcal{S}_\mathcal{B})$.

Also, the time complexities we provided here are rather loose. In reality, input data is often very sparse, and it turns out that the $|\mathcal{V}|^n$ term in the time complexities can be significantly reduced. See appendix for more discussion.

## 4.4 Learning

In this section, we discuss how we solve the two problems stated in Sec. 3, i.e., mining logic rules and learning the probabilistic prediction model.

**Learning Symbolic Structures via Continuous Optimization** We now discuss how we learn the structures of logic classifiers, i.e., to solve the problem

$$\max_{\varphi} \quad p(Q(\mathbf{x}) \in \mathcal{S}_\mathcal{P}|\varphi(\mathbf{x}) = 1). \tag{11}$$

**Theorem 4.4** (Sparse attentions). *Consider the optimization problem*

$$\min_{\psi} \quad \mathcal{L}(\psi) = -\log \mathbb{E}_{\mathbf{x} \sim p_1}[\psi(\mathbf{x}; \alpha)] + \log \mathbb{E}_{\mathbf{x} \sim p_2}[\psi(\mathbf{x}; \beta)], \tag{12}$$

*where $\psi(\mathbf{x}; \alpha)$ and $\psi(\mathbf{x}; \beta)$ are obtained by one iteration of the inference procedure. Here, we assume $\mathbb{E}_{\mathbf{x} \sim p_1}[\psi(\mathbf{x}; \alpha)] > 0$ and $\mathbb{E}_{\mathbf{x} \sim p_2}[\psi(\mathbf{x}; \beta)] > 0$. With the following constraints being satisfied: (1) $\alpha > \beta > 1$; (2) $\psi$ is the root of a LP-tree; (3) Negations are applied only on leaf nodes. Then, at each local minima of $\mathcal{L}(\psi)$, the attention vectors $\mathbf{w}$, $\mathbf{w}^{(i)}, ...$ used for evaluating $\psi$ are one-hot vectors and $\psi$ explicitly captures a logic classifier.*

We say $\psi$ captures a logic classifier $\varphi$ when evaluated on any background assumption $\mathcal{S}_\mathcal{B}$, $\psi(\mathbf{x}) > 0 \iff \varphi(\mathbf{x}) = 1$ for any $\mathbf{x}$. With the above theorem, we can directly derive the following corollary.

**Corollary 4.5** (Proxy problem). *Minimization of the optimization problem*

$$\min_{\psi} \quad \mathcal{L}(\psi) = -\log \mathbb{E}\left[\psi(\mathbf{x}; \alpha)|Q(\mathbf{x}) \in \mathcal{S}_\mathcal{P}\right] + \log \mathbb{E}\left[\psi(\mathbf{x}; \beta)|Q(\mathbf{x}) \in \mathcal{S}_\mathcal{N}\right], \tag{13}$$

*yields a near-optimal solution for solving problem 11, with the constraints of theorem 4.4 being satisfied.*

We say the solution is near-optimal because (1) the LPs are only capable to capture the logic classifiers within their expressiveness power, and (2) the ranges of $\varphi$ and $\psi$ are different: we have $\varphi(\mathbf{x}) \in \{0, 1\}$ and $\psi(\mathbf{x}; \alpha) \in [0, +\infty)$. For example, if we have $\varphi(x) := \exists y : \mathrm{Neighbor}(x, y)$, then the corresponding $\psi(x)$ is equal to the amount of neighbors that $x$ have. Also, the 3 constraints are sufficient conditions for the optimization problem to guarantee to converge at the points where $\psi$ captures a logic classifier, but they are not always necessary. In most cases we can relax these constraints and set $\alpha = \beta = 1$ while keeping the sparsity of the solutions, which is shown in the experiments. The LP-tree constructed from the grammar 1 naturally satisfies the constraint (2), while for LP-layer the constraint (2) is relaxed. See appendix C for more discussion of how these constraints work.

**Learning the Inference Model**   To learn the inference model $p(Q(\mathbf{x}) \in \mathcal{S}_\mathcal{P}|\mathcal{S}_\mathcal{B})$, we fix the parameters of each $\psi \in \Psi$. Then, we proceed the inference steps for a fixed number $T$. Since the Update function is differentiable w.r.t. its parameters, the obtained $p(Q(\mathbf{x}) \in \mathcal{S}_\mathcal{P}|\mathcal{S}_\mathcal{B})$ for each $Q \in \mathcal{P}$ and $\mathbf{x}$ is also differentiable, and we can optimize the inference model by simply maximizing the likelihood of $p(Q(\mathbf{x}) \in \mathcal{S}_\mathcal{P}|\mathcal{S}_\mathcal{B})$ on data.

With the proposed approaches, the whole learning procedure of our model described in Sec. 4.1 is realized.

# 5   EXPERIMENT

## 5.1   DATASETS

We consider datasets from different fields to test the model's ability in solving ILP tasks, systematic reasoning and knowledge graph completion. These datasets include:

**ILP tasks**   We test the model's expressiveness by applying the model to solve the 20 ILP tasks proposed in Evans & Grefenstette (2018). These tasks test the expressive power of an ILP model, including learning the concepts of even numbers, family relations, graph coloring, etc.

**Systematicity**   We test the model's systematicity (Lu et al. (2022)) on CLUTRR (Sinha et al. (2019)) datasets. These tasks test a model's ability of generalizing on unseen data of different distributions. Models are trained on small scale of data and tested on larger data with longer resolution steps.

**Knowledge graph completion**   We test the model's capability of performing probabilistic reasoning on knowledge graphs including UMLS, Kinship (Kok & Domingos (2007)), WN18RR (Dettmers et al. (2018)) and FB15k-237 (Toutanova & Chen (2015)). These tasks test a model's ability of dealing with probabilistic and noisy data. For Kinship and UMLS, since there are no standard data splits, we take the data splits from Qu et al. (2021).

## 5.2   MODEL CONFIGURATION

On all experiments, we use the grammar presented in Eq. 1, and if the input data does not contain unary predicates, we simply use an invented one $P_{inv}(x) \equiv 1$. We stack LPs the same way provided in Sec. 4.2, where the depth of LP-tree and number of layers of LP-layer are 5 for ILP tasks, 3 for Systematicity tasks and KG completion. For inference model, the number of iterations is 10 for ILP and Systematicity tasks and 1 for KG completion. Due to space constraints, we left the detailed model configuration and a sketch figure of the constructed network in Appendix D.

## 5.3   COMPARED ALGORITHMS

We observe there are few models capable of solving all the tasks, so we pick different algorithms for comparison for different tasks. For ILP tasks, we choose Evans & Grefenstette (2018). For systematic reasoning, we choose Graph Attention Networks (GAT) (Velickovic et al. (2018)), Graph Convolutional Networks (GCN) (Kipf & Welling (2017)), Recurrent Neural Networks (RNN) (Schuster & Paliwal (1997)), Long Short-Term Memory Networks (LSTM) (Hochreiter & Schmidhuber (1997)), Gated Recurrent Units (GRU) (Cho et al. (2014)), Convolutional Neural Networks (CNN) (Kim (2014)), CNN with Highway Encoders (CNNH) (Kim et al. (2016)), Greedy Neural Theorem Provers (GNTP) (Minervini et al. (2020a)), Multi-Headed Attention Networks (MHA) (Vaswani et al. (2017)), Conditional Theorem Provers (CTP) (Minervini et al. (2020b)), R5 (Lu et al. (2022)). For knowledge graph completion, we choose rule-based methods including Markov Logic Networks (MLN) (Richardson & Domingos (2006)), PathRank (Lao & Cohen (2010)), NeuralLP (Yang et al. (2017)), DRUM (Sadeghian et al. (2019)), CTP (Minervini et al. (2020b)), M-Walk (Shen et al. (2018)), MINERVA (Das et al. (2018)), NLIL (Yang & Song (2020)), RNNLogic (Qu et al. (2021)).

Table 1: Results on CLUTRR.

| Method | Short Stories | | | | | | | Long Stories | | | | | |
|---|---|---|---|---|---|---|---|---|---|---|---|---|---|
| | 4Hops | 5Hops | 6Hops | 7Hops | 8Hops | 9Hops | 10Hops | 5Hops | 6Hops | 7Hops | 8Hops | 9Hops | 10Hops |
| DLP-tree | .990±.006 | **.994±.001** | **1.0±.000** | **.995±.002** | **.997±.001** | **.996±.000** | **1.0±.000** | **.992±.001** | .990±.000 | .994±.002 | **1.0±.000** | **.992±.002** | **.996±.001** |
| DLP-layer | **.991±.003** | .993±.001 | **1.0±.000** | **.995±.002** | **.997±.000** | **.996±.000** | **1.0±.000** | **.992±.002** | .990±.000 | **.997±.001** | **1.0±.000** | **.992±.002** | **.996±.001** |
| R5 | .98±.02 | **.99±.02** | .98±.03 | .96±.05 | .97±.01 | .98±.03 | .97±.03 | .99±.02 | .99±.04 | **.99±.03** | **1.0±.02** | **.99±.02** | .98±.03 |
| CTP$_L$ | .98±.02 | .98±.03 | .97±.05 | .96±.04 | .94±.05 | .89±.07 | .89±.07 | .99±.02 | .98±.04 | .97±.04 | .98±.03 | .97±.04 | .95±.04 |
| CTP$_A$ | .99±.02 | .99±.01 | .99±.02 | .96±.04 | .94±.05 | .89±.08 | .90±.07 | .99±.04 | .99±.03 | .97±.03 | .95±.06 | .93±.07 | .91±.05 |
| CTP$_M$ | .97±.03 | .97±.03 | .96±.06 | .95±.06 | .93±.05 | .90±.06 | .89±.06 | .98±.04 | .97±.06 | .95±.06 | .94±.08 | .93±.08 | .90±.09 |
| GNTP | .49±.18 | .45±.21 | .38±.23 | .37±.21 | .32±.20 | .31±.19 | .31±.22 | .68±.28 | .63±.34 | .62±.31 | .59±.32 | .57±.34 | .52±.32 |
| GAT$_s$ | .91±.02 | .76±.06 | .54±.03 | .56±.04 | .54±.03 | .55±.05 | .45±.06 | .99±.00 | .85±.04 | .80±.03 | .71±.03 | .70±.03 | .68±.02 |
| GCN$_s$ | .84±.03 | .68±.02 | .53±.03 | .47±.04 | .42±.03 | .45±.03 | .39±.02 | .94±.03 | .79±.02 | .61±.03 | .53±.04 | .53±.04 | .41±.04 |
| RNN$_s$ | .86±.06 | .76±.08 | .67±.08 | .66±.08 | .56±.10 | .55±.10 | .48±.07 | .93±.06 | .87±.07 | .79±.11 | .73±.12 | .65±.16 | .64±.16 |
| LSTM$_s$ | .98±.04 | .95±.03 | .88±.05 | .87±.04 | .81±.07 | .75±.10 | .75±.09 | .98±.03 | .95±.04 | .89±.10 | .84±.07 | .77±.11 | .78±.11 |
| GRU$_s$ | .89±.05 | .83±.06 | .74±.12 | .72±.09 | .67±.12 | .62±.10 | .60±.12 | .95±.04 | .94±.03 | .87±.08 | .81±.13 | .74±.15 | .75±.15 |
| CNNH$_s$ | .90±.04 | .81±.05 | .69±.10 | .64±.08 | .56±.13 | .52±.12 | .50±.12 | .99±.01 | .97±.02 | .94±.03 | .88±.04 | .86±.05 | .84±.06 |
| CNN$_s$ | .95±.02 | .90±.03 | .89±.04 | .80±.05 | .76±.08 | .69±.07 | .70±.08 | **1.0±.00** | **1.0±.01** | .98±.01 | .95±.03 | .93±.03 | .92±.04 |
| MHA$_s$ | .81±.04 | .76±.04 | .74±.05 | .70±.04 | .69±.03 | .64±.05 | .67±.02 | .88±.03 | .83±.05 | .76±.04 | .72±.04 | .74±.05 | .70±.03 |

## 5.4 RESULTS

**1. Comparison with other methods.** The main results of systematicity tests and KG completion are shown in Tab. 1, Tab. 2 and Tab. 3. The 20 ILP tasks are pass-or-fail tests and both LP-tree and LP-layer are able to solve all of them, indicating that our model has sufficient expressive power to learn a variety of general ILP problems. In contrast to $\partial$ILP (Evans & Grefenstette (2018)), a principle ILP method which uses different program templates to solve the problems, our approach uses the same grammar for all problems. With the same model architecture, our model is able to achieve high accuracy on the CLUTRR datasets. Our model is also able to learn valuable logic rules and make fairly accurate predictions on much noisier knowledge graphs.

**2. Performance w.r.t. rule complexity.** We conduct experiments to study the model performance under different rule complexity on UMLS dataset in Tab. 4. We can see that if we force the rules to be too simple, it's hard to capture informative patterns of data; On the other hand, if we force the rules to be too complex, the performance is also decreased due to the loss of rule generality.

**3. Performance w.r.t. reweighting techniques.** We study the effects of reweighting training data in Tab. 5. We train models on UML dataset with different reweighting methods. We can see that even without reweighting, our model is able to capture various logical patterns by merely randomly initializing model parameters. Besides, replacing, i.e., removing data instances that are correctly predicted, achieved worst results. This is because for noisy data it's better to learn more different logical patterns to prove the targets, and removing them prevents the model to learn more information about them.

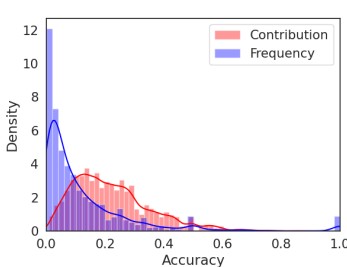

Figure 2: Distribution of rules.

**4. Effects of fine-tuning rule weights.** We find on most situations the model is able to make fairly precise predictions without training rule weights as in Tab. 6. We set the weights of each rule to be its precision on training data, and conduct comparison experiments based on that.

**5. Effects of hyperparameters $\alpha$ and $\beta$.** To show that in most situations we can safely set $\alpha$ and $\beta$ to a relative small value, we conduct experiments on UMLS dataset with different settings of $\alpha$ and $\beta$, summarized in Tab. 7. Suprisingly, the performance under different $\alpha$ and $\beta$ greater or equal than 1.0 is quite similar. This implies that on most situations we can safely set $\alpha$ and $\beta$ to 1.0 to both simplify computation and stabilize the learning of model.

**6. Distribution of rules accuracy.** We summarize the distributions of learnt rule accuracy and average rule contributions for UMLS in Fig. 2. The average contribution of a rule equals to the average decrement of scores of queries in test data if we remove the rule and hence it measures how important a rule is on average for predicting the correct answers. We can see that most learnt rule

Table 2: Results on Kinship and UMLS.

| Method | Kinship | | | | | UMLS | | | | |
|---|---|---|---|---|---|---|---|---|---|---|
| | MR | MRR | H@1 | H@3 | H@10 | MR | MRR | H@1 | H@3 | H@10 |
| MLN | 10.0 | 0.351 | 0.189 | 0.408 | 0.707 | 7.6 | 0.688 | 0.587 | 0.755 | 0.869 |
| PathRank | - | 0.369 | 0.272 | 0.416 | 0.673 | - | 0.197 | 0.148 | 0 214 | 0.252 |
| NeuralLP | 16.9 | 0.302 | 0.167 | 0.339 | 0.596 | 10.3 | 0.483 | 0.332 | 0.563 | 0.775 |
| DRUM | 11.6 | 0.334 | 0.183 | 0.378 | 0.675 | 8.4 | 0.548 | 0.358 | 0.699 | 0.854 |
| MINERVA | - | 0.401 | 0.235 | 0.467 | 0.766 | - | 0.564 | 0.426 | 0.658 | 0.814 |
| CTP | - | 0.335 | 0.177 | 0.376 | 0.703 | - | 0.404 | 0.288 | 0.430 | 0.674 |
| RNNLogic (w/o emb.) | 3.9 | 0.639 | 0.495 | 0.731 | 0.924 | 5.3 | 0.745 | 0.630 | 0.833 | 0.924 |
| DLP | **3.7** | **0.645** | **0.504** | **0.733** | **0.927** | **3.1** | **0.810** | **0.708** | **0.896** | **0.959** |

Table 3: Results on FB15k-237 and WN18RR.

| Method | FB15k-237 | | | | | WN18RR | | | | |
|---|---|---|---|---|---|---|---|---|---|---|
| | MR | MRR | H@1 | H@3 | H@10 | MR | MRR | H@1 | H@3 | H@10 |
| PathRank | - | 0.087 | 0.074 | 0.092 | 0.112 | - | 0.189 | 0.171 | 0.200 | 0.225 |
| NeuralLP | - | 0.237 | 0.173 | 0.259 | 0.361 | - | 0.381 | 0.368 | 0.386 | 0.408 |
| DRUM | - | 0.238 | 0.174 | 0.261 | 0.364 | - | 0.382 | 0.369 | 0.388 | 0.410 |
| NLIL | - | 0.25 | - | - | 0.324 | - | - | - | - | - |
| M-Walk | - | 0.232 | 0.165 | 0.243 | - | - | 0.437 | 0.414 | 0.445 | - |
| RNNLogic (w/o emb.) | 538 | 0.288 | 0.208 | 0.315 | 0.445 | 7527 | 0.455 | 0.414 | 0.475 | 0.531 |
| RNNLogic+ (w/o emb.) | 480 | **0.299** | **0.215** | **0.328** | **0.464** | 7204 | 0.489 | 0.453 | 0.506 | **0.563** |
| DLP | **432** | 0.285 | 0.208 | 0.310 | 0.436 | **7190** | **0.501** | **0.472** | **0.514** | 0.556 |

are rather inaccurate, as there are barely any rules having precision higher than $0.5$, but putting them together, rules of accuracy $0 - 0.4$ contribute the most for proving the target queries.

Table 4: Study of rule complexity.

| Rule Length | 2 | 3 | 4 |
|---|---|---|---|
| MRR | 0.682 | 0.810 | 0.786 |

Table 5: Study of reweighting.

| Methods | Reweight | Replace | None |
|---|---|---|---|
| MRR | 0.810 | 0.584 | 0.733 |

Table 6: Study of fine-tuning.

| Fine-tune | Y | N |
|---|---|---|
| MRR | 0.810 | 0.802 |

Table 7: Study of $\alpha$ and $\beta$.

| $\alpha$ and $\beta$ | 0.5 | 1.0 | 2.0 | 3.0 |
|---|---|---|---|---|
| MRR | 0.687 | 0.810 | 0.809 | 0.802 |

## 6 CONCLUSION

This paper studies inductive logic programming, and we propose Differentiable Logic Programming framework to solve the problems of structure learning and weights learning. We generalize the discrete rule search problem and forward chaining algorithm in a continuous and probabilistic manner and use a differentiable program with a proxy problem to solve the learning problem. Both theoretical and empirical evidences are present to prove the efficiency of our algorithm. In the future, we plan to explore the possibility of combining neural network architecture to help the model discovery more complex and accurate logical patterns.

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

## A    PROOF OF PROPOSITION 4.2

We now prove Prop. 4.2.

**Proposition A.1** (Properties of $\Phi$). *Given the input data $(\mathcal{S_B}, \mathcal{S_P}, \mathcal{S_N})$, let $Q$ be a d-ary target predicate, $\Phi_Q = \{\varphi_0, \varphi_1, ..., \varphi_{L-1}\}$ be a set of logic classifiers and $p_0 \in (0, 1]$ a fixed threshold. With the following statements being satisfied: (1) We start from $l = 0$ and let $\mathcal{S_P}^{(0)} = \mathcal{S_P}$; (2) For $\varphi_l$, its precision: $p(Q(\mathbf{x}) \in \mathcal{S_P}^{(l)}|\varphi_l(\mathbf{x}) = 1) = \frac{\sum_{\mathbf{x} \in \mathcal{V}^d} \varphi_l(\mathbf{x}) \mathbf{1}_{Q(\mathbf{x}) \in \mathcal{S_P}^{(l)}}}{\sum_{\mathbf{x} \in \mathcal{V}^d} \varphi_l(\mathbf{x}) \mathbf{1}_{Q(\mathbf{x}) \in \mathcal{S}_P^{(l)} \cup \mathcal{S_N}}} \geq p_0$; (3) We let $\mathcal{S_P}^{(l+1)} = \mathcal{S_P}^{(l)} \setminus \{Q(\mathbf{x}) \mid \mathbf{x} \in \mathcal{V}^d, \varphi_l(\mathbf{x}) = 1\}$ and increase $l$ by 1 and go to (2) again until $l = L$; (4) $\mathcal{S_P}^{(L)} = \emptyset$. Then, there exists a prediction model $p(Q(\mathbf{x}) \in \mathcal{S_P}|\mathcal{S_B}) = f(\varphi_0(\mathbf{x}), \varphi_1(\mathbf{x}), ..., \varphi_{L-1}(\mathbf{x}))$ such that its error rate satisfies:*

$$\text{Err}\,[f; (\mathcal{S_B}, \mathcal{S_P}, \mathcal{S_N})] \leq \frac{(1-p_0)N_Q}{p_0 N} \leq 1 - p_0, \tag{14}$$

*where $N_Q = \sum_{\mathbf{x} \in \mathcal{V}^d} \mathbf{1}_{Q(\mathbf{x}) \in \mathcal{S_P}}$ and $N = \sum_{\mathbf{x} \in \mathcal{V}^d} \mathbf{1}_{Q(\mathbf{x}) \in \mathcal{S_P} \cup \mathcal{S_N}}$.*

**Proof:**   The proof is constructive. We let $f(\varphi_0(\mathbf{x}), ..., \varphi_{L-1}(\mathbf{x})) = \min\left\{1, \sum_{\varphi \in \Phi} \varphi(\mathbf{x})\right\}$, and let $N_{\varphi_i}$ be the number of $Q(\mathbf{x})$s that are true and also predicted true by $\varphi_i$, with those $Q(\mathbf{x})$s related to $\varphi_1, \varphi_2, ..., \varphi_{i-1}$ removed first. It's easy to observe that the number of $Q(\mathbf{x})$s predicted true by $\varphi_i$ but are actually false is $W_{\varphi_i} \leq \frac{1-p}{p} N_{\varphi_i}$. For simplification here we let $Q(\mathbf{x}) = 1$ if $Q(\mathbf{x}) \in \mathcal{S_P}$ and 0 otherwise. Thus, we have:

$$\begin{aligned}
\text{Err}\,[f; (\mathcal{S_B}, \mathcal{S_P}, \mathcal{S_N})] &= \frac{\sum_{\mathbf{x}} \mathbf{1}_{Q(\mathbf{x}) \neq p_w(Q(\mathbf{x}))}}{N} \\
&= \frac{1}{N_{\mathcal{G}}} \sum_{\mathbf{x}} (1 - Q(\mathbf{x})) \min\left\{1, \sum_{\varphi \in \Phi} \varphi(\mathbf{x})\right\} \\
&\leq \frac{1}{N} \sum_i W_{\varphi_i} \\
&\leq \frac{1}{N} \sum_i \frac{1-p}{p} N_{\varphi_i} \\
&= \frac{(1-p)N_Q}{pN} \leq 1 - p.
\end{aligned} \tag{15}$$

## B    PROOF OF THEOREM 4.4 AND COROLLARY 4.5

We now prove Prop. 4.5 and Corollary 4.5 .

**Theorem B.1** *Consider the optimization problem*

$$\min_{\psi} \quad \mathcal{L}(\psi) = -\log \mathbb{E}_{\mathbf{x} \sim p_1}[\psi(\mathbf{x}; \alpha)] + \log \mathbb{E}_{\mathbf{x} \sim p_2}[\psi(\mathbf{x}; \beta)], \tag{16}$$

*where $\psi(\mathbf{x}; \alpha)$ and $\psi(\mathbf{x}; \beta)$ are obtained by one iteration of the inference procedure. Here, we assume $\mathbb{E}_{\mathbf{x} \sim p_1}[\psi(\mathbf{x}; \alpha)] > 0$ and $\mathbb{E}_{\mathbf{x} \sim p_2}[\psi(\mathbf{x}; \beta)] > 0$. With the following constraints being satisfied: (1) $\alpha > \beta > 1$; (2) $\psi$ is the root of a LP-tree; (3) Negations are applied only on leaf nodes. Then, at each local minima of $\mathcal{L}(\psi)$, the attention vectors $\mathbf{w}$ used for evaluating $\psi$ are one-hot vectors and $\psi$ explicitly captures a logic classifier.*

**Proof:**   Before we proceed to prove the proposition, we need to first study in what situations does a LP $\psi$ capture a logic classifier $\varphi$. Across this section we assume we are given a inference network $\mathcal{G}$ composed of $\{\psi_1, \psi_2, ..., \psi_L, \psi\}$ where $\psi$ is the output node built upon $\{\psi_1, \psi_2, ..., \psi_L\}$ we care about. The following lemma explains the properties of $\psi$ when it captures a logic classifier.

**Lemma B.2** *Given $\mathcal{G}$ and $\{\psi_1, \psi_2, ...\}$ stated above. If we perform a restricted breadth first search across the inference network where:*

- *We start with the output node $\psi$;*

- *For each node we go through, the parameters of the node $\boldsymbol{w}$ defined in Eq. 6-7 are a one-hot vector where one of its dimensions equals to $1$;*

- *From the current node, the next paths we go are a subset of the inverse edges of $\psi_i$, where we only consider the ones corresponding to the nonzero dimensions of $\boldsymbol{w}$ discussed above defined in Eq. 6-7.*

*Then, $\psi$ captures a logic classifier $\varphi$, i.e., for all $\mathbf{x}$, $\psi(\mathbf{x}) > 0 \iff \varphi(\mathbf{x}) > 0$.*

The lemma is quite straightforward: by restricting a LP with the above constraints, its evaluation procedure naturally simulates the grounding of a logic classifier. We can construct a logic classifier $\varphi$ captured by $\psi$ as follows: for each node we went through in the procedure of lemma B.2, we replace the LP $\psi$ with an actual logic classifier $\varphi$ which is constructed with the correspondence defined in Eq. 7. Moreover, for each one-hot vectors $\boldsymbol{w}$, we pick the predicates or sub-formulas corresponding to the nonzero dimension of these vectors. By doing so, we observe that the evaluation results of $\varphi$ for any $\mathbf{x}$ are the same as $\psi$ does because every computational steps of both $\varphi$ and $\psi$ stay the same. Thus, to prove that a $\psi$ captures a $\varphi$, we need to show that the parameters of $\psi$ satisfy the constraints in lemma B.2.

We now proceed to prove Theorem 4.4 recursively: we first prove that the parameters $\boldsymbol{w}$ of $\psi$ must satisfies the proposition, i.e. it is one-hot, no matter whether $\boldsymbol{w}^{(1)}$, $\boldsymbol{w}^{(2)}$, ... of $\mathcal{F}_1$, $\mathcal{F}_2$,... of $\psi$ are or not; Then, we prove that the parameters $\boldsymbol{w}^{(i_1)}$, $\boldsymbol{w}^{(i_2)}$, ... w.r.t. the nonzero dimension $w$ of $\boldsymbol{w}$ must also be one-hot; After that, we show that if $\psi$ satisfies the proposition, the next nodes of $\psi$ in the restricted BFS path must also satisfies the proposition; Hence the theorem is proved.

To study $\boldsymbol{w}$, we fix all other parameters in the inference network $\mathcal{G}$ except $\boldsymbol{w}$ for $\psi$, and the evaluation of $\psi(\mathbf{x}; \alpha)$ becomes a simple polynomial function:

$$\psi(\mathbf{x}; \alpha) = \sum_i \mathcal{F}_i(\mathbf{x}) w_i^\alpha, \tag{17}$$

and the evaluation of $\mathcal{L}'$ becomes:

$$
\begin{aligned}
\mathcal{L}'(\psi) &= -\log \mathbb{E}_{\mathbf{x} \sim p}[\psi(\mathbf{x}; \alpha)] + \log \mathbb{E}_{\mathbf{x} \sim q}[\psi(\mathbf{x}; \beta)] \\
&= -\log \sum_{\mathbf{x}} \psi(\mathbf{x}; \alpha) p(\mathbf{x}) + \log \sum_{\mathbf{x}} \psi(\mathbf{x}; \beta) q(\mathbf{x}) \\
&= -\log \left\{ \left( \sum_{\mathbf{x}} \sum_i p(\mathbf{x}) \mathcal{F}_i(\mathbf{x}; \alpha) \right) w_i^\alpha \right\} + \log \left\{ \left( \sum_{\mathbf{x}} \sum_i q(\mathbf{x}) \mathcal{F}_i(\mathbf{x}; \beta) \right) w_i^\beta \right\}.
\end{aligned}
\tag{18}
$$

Hence, $\mathcal{L}'(\psi)$ is a polynomial function w.r.t. $\boldsymbol{w}$ and we need to show that at each local minima of $\mathcal{L}'(\psi)$ there cannot exist two $w_i$ and $w_j$ that are all larger than 0, which will be discussed later in this section. Now, assume we have already know that $\boldsymbol{w}$ of $\psi$ satisfies the proposition, i.e., $w_i = 1$ for some $i$. To proceed, we need to prove that all parameters $\boldsymbol{w}^{(i_1)}$, $\boldsymbol{w}^{(i_2)}$, ... w.r.t. $\mathcal{F}_i$ are also one-hot. We do this similarly as before: we fix all other parameters in the inference network except some $\boldsymbol{w}^{(i_k)}$ emerged in a neighbor of $\psi$ in the inference network. By carefully checking the construction steps of $\mathcal{F}_i$ in Eq. 7, it's easy to show that $\mathcal{L}'(\psi)$ is also a polynomial function w.r.t. $\boldsymbol{w}^{(i_k)}$ under the constraints that $\boldsymbol{w}$ is one-hot. The following theorem directly proves that these functions indeed satisfy the constraints.

**Theorem B.3** (Sparse Attentions.) *The following function*

$$h(\boldsymbol{w}, \alpha, \beta) = \frac{\sum_i A_i(\alpha) w_i^\alpha}{\sum_i B_i(\beta) w_i^\beta} \tag{19}$$

*has no local maxima w.r.t. $w_i \in (0, 1)$ with the following constraints being satisfied:*
*(1) Attention vector: $w_i \geq 0$ for each dimension $i$ and $\sum_i w_i = 1$;*

*(2) Positive coefficients:* $A_i(\alpha) \geq 0$ and $B_i(\beta) \geq 0$ *for every* $i$ *and all* $\alpha, \beta > 0$;
*(3) Non-empty results:* $\sum_i A_i(\alpha)w_i^{\alpha} > 0$ and $\sum_i B_i(\beta)w_i^{\beta} > 0$;
*(4)* $\alpha > \beta > 1$.

**Proof of theorem B.3:** We now prove that $h(\boldsymbol{w}, \alpha, \beta)$ has no local maxima where each $w_i \in (0, 1)$ by contradiction. Assume we are now given some $\boldsymbol{w}_0$ where there are at least two dimensions of $\boldsymbol{w}_0$ that are nonzero, and the target is to prove that $\boldsymbol{w}_0$ is not a local maxima of $h$. To do so, we create a new vector $\boldsymbol{v}_0$ composed of nonzero dimensions of $\boldsymbol{w}_0$ together with a function $h'(\boldsymbol{v}, \alpha, \beta)$, and

$$h(\boldsymbol{w}, \alpha, \beta) = h'(\boldsymbol{v}, \alpha, \beta) = \frac{\sum_i C_i(\alpha)v_i^{\alpha}}{\sum_i D_i(\beta)v_i^{\beta}} = \frac{f(\boldsymbol{v}, \alpha)}{g(\boldsymbol{v}, \beta)}. \tag{20}$$

To prove that $h(\boldsymbol{w}, \alpha, \beta)$ is not at local maxima, we can instead show that $\boldsymbol{v}_0$ is not a local maxima of $h'(\boldsymbol{v}, \alpha, \beta)$ is not at local maxima. We first consider the situations where the partial derivatives of $h'$ w.r.t. each $v_i$ are not all the same, for example, suppose $\frac{\partial h'}{\partial v_1}|_{\boldsymbol{v}=\boldsymbol{v}_0} > \frac{\partial h'}{\partial v_2}|_{\boldsymbol{v}=\boldsymbol{v}_0}$. Directly study all dimensions of $\boldsymbol{v}$ is intractable with constraint $\sum_i v_i = 1$, so we instead fix all parameters $v_3, v_4, \ldots$ at the corresponding value of $\boldsymbol{v}_0$ and only study the two parameters $v_1$ and $v_2$ where we let $v_1 = v_{01} + x$, $v_2 = v_{02} - x$ and so the above constraint is naturally satisfied. Thus, we have $h'(\boldsymbol{v}, \alpha, \beta) = h''(x)$ and

$$\frac{dh''(x)}{dx}\bigg|_{x=0} = \frac{\partial h'(\boldsymbol{v}, \alpha, \beta)}{\partial v_1}\bigg|_{\boldsymbol{v}=\boldsymbol{v}_0} - \frac{\partial h'(\boldsymbol{v}, \alpha, \beta)}{\partial v_2}\bigg|_{\boldsymbol{v}=\boldsymbol{v}_0} > 0. \tag{21}$$

Since the derivative w.r.t. $x$ is larger than 0, $\boldsymbol{v}_0$ is not a local maxima.

Next, we discuss the situation where $\frac{\partial h'}{\partial v_i}|_{\boldsymbol{v}=\boldsymbol{v}_0} = \lambda$ are all the same. We again let $v_1 = v_{01} + x$, $v_2 = v_{02} - x$, and fix all other parameters of $\boldsymbol{v}$. We have

$$\begin{aligned}
\frac{d^2 h''(x)}{dx^2}\bigg|_{x=0} = \frac{1}{g^3}\bigg\{ & g^2\alpha(\alpha - 1)\left(A_1 v_1^{\alpha-2} + A_2 v_2^{\alpha-2}\right) \\
& - 2g\alpha\beta(A_1 v_1^{\alpha-1} + A_2 v_2^{\alpha-1})(B_1 v_1^{\beta-1} + B_2 v_2^{\beta-2}) \\
& + 2f\beta^2\left(B_1 v_1^{\beta-1} + B_2 v_2^{\beta-2}\right)^2 \\
& - gf\beta(\beta - 1)\left(B_1 v_1^{\beta-2} + B_2 v_2^{\beta-2}\right)\bigg\}.
\end{aligned} \tag{22}$$

Since we assume $\frac{\partial h'}{\partial v_i}|_{\boldsymbol{v}=\boldsymbol{v}_0} = \lambda$, we have

$$\begin{aligned}
& \frac{\partial h'}{\partial v_i}\bigg|_{\boldsymbol{v}=\boldsymbol{v}_0} = \frac{1}{g^2}\left(\alpha g A_i v_i^{\alpha-1} - \beta f B_i v_i^{\beta-1}\right) = \lambda, \\
\implies & \frac{1}{g^2}\left(\alpha g A_i v_i^{\alpha} - \beta f B_i v_i^{\beta}\right) = v_i \lambda, \\
\implies & \sum_i \frac{1}{g^2}\left(\alpha g A_i v_i^{\alpha} - \beta f B_i v_i^{\beta}\right) = \sum_i v_i \lambda, \\
\implies & \frac{f}{g}(\alpha - \beta) = \lambda.
\end{aligned} \tag{23}$$

Substituting Eq. 23 into Eq. 22, we have

$$
\begin{aligned}
\frac{d^2 h''(x)}{dx^2}\bigg|_{x=0} &= \frac{f}{g}\left(\frac{1}{f}A_1\alpha(\alpha-1)v_1^{\alpha-2} - \frac{1}{g}B_1\beta(\beta-1)v_1^{\beta-2}\right) \\
&+ \frac{f}{g}\left(\frac{1}{f}A_2\alpha(\alpha-1)v_2^{\alpha-2} - \frac{1}{g}B_2\beta(\beta-1)v_2^{\beta-2}\right) \\
&\geq \frac{f}{gv_1}\left(\frac{1}{f}A_1\alpha v_1^{\alpha-1} - \frac{1}{g}B_1\beta v_1^{\beta-1}\right) + \frac{f}{gv_2}\left(\frac{1}{f}A_2\alpha v_2^{\alpha-1} - \frac{1}{g}B_2\beta v_2^{\beta-1}\right) \quad (24) \\
&= \frac{\lambda}{v_1} + \frac{\lambda}{v_2} \\
&= \frac{f}{g}(\alpha-\beta)\left(\frac{1}{v_1} + \frac{1}{v_2}\right) > 0.
\end{aligned}
$$

So $\boldsymbol{v}_0$ is not a local maxima of $h'$. Thus, we have proved that $h(\boldsymbol{w}, \alpha, \beta)$ has no local maxima for $w_i \in (0, 1)$.

**End of proof of theorem B.3.**

Since $f$ and $\log f$ share the same minimum points for any $f > 0$ we have shown that the output node $\psi$ indeed satisfies the conditions in the proposition. It's straightforward to show the nodes along the paths that $\psi$ is built on also satisfy the conditions. Suppose we are studying another $\psi'$ that is used for computing $\psi$. By writing the detailed computation steps for evaluating $\psi$ and fixing all irrelevant parameters, we can show that $\psi(\mathbf{x}; \alpha) = \sum_{\mathbf{x}'} A(\mathbf{x}', \alpha)\psi(\mathbf{x}'; \alpha)$ and thus

$$
\mathbb{E}_{x \sim p}[\psi(\mathbf{x}; \alpha)] = \mathbb{E}_{x' \sim p'}[\psi(\mathbf{x}'; \alpha)], \quad (25)
$$

where $p'$ is an unnormalized distribution. Thus, the same conclusion holds for all $\psi$ in the restricted BFS path.

We now have proved that every local minima of the proxy problem corresponds to a logic classifier, and it's much easier to prove the rest of the conclusions as collaborations of the first one. For conclusion (2), we notice that when $\psi$ converges,

$$
\begin{aligned}
p(Q(\mathbf{x})|\varphi(\mathbf{x}) = 1) &= \frac{p(\varphi(\mathbf{x}) = 1|Q(\mathbf{x}) = 1)}{p(\varphi(\mathbf{x}) = 1)} p(Q(\mathbf{x}) = 1) \\
&= \frac{\mathbb{E}[\varphi(\mathbf{x})|Q(\mathbf{x}) = 1]}{\mathbb{E}[\varphi(\mathbf{x})]} \mathbb{E}[Q(\mathbf{x})] \\
&\approx \frac{\mathbb{E}[\psi(\mathbf{x}; \alpha)|Q(\mathbf{x}) = 1]}{\mathbb{E}[\psi(\mathbf{x}; \beta)]} \frac{N_Q}{N} \\
&= \frac{\mathbb{E}[\psi(\mathbf{x}; \alpha)|Q(\mathbf{x}) = 1]}{\mathbb{E}[\psi(\mathbf{x}; \beta)|Q(\mathbf{x}) \neq 1] + \mathbb{E}[\psi(\mathbf{x}; \beta)|Q(\mathbf{x}) = 1]} \frac{N_Q}{N} \quad (26) \\
&= \frac{\mathbb{E}[\psi(\mathbf{x}; \alpha)|Q(\mathbf{x}) = 1]}{\mathbb{E}[\psi(\mathbf{x}; \beta)|Q(\mathbf{x}) \neq 1] + \mathbb{E}[\psi(\mathbf{x}; \alpha)|Q(\mathbf{x}) = 1]} \frac{N_Q}{N} \\
&= \frac{1}{1 + \frac{\mathbb{E}[\psi(\mathbf{x}; \beta)|Q(\mathbf{x}) \neq 1]}{\mathbb{E}[\psi(\mathbf{x}; \alpha)|Q(\mathbf{x}) = 1]}} \frac{N_Q}{N} \\
&= \frac{N_Q}{(1 + \exp \mathcal{L}(\psi))N},
\end{aligned}
$$

which is monotone decreasing w.r.t. $\mathcal{L}(\psi)$ so conclusion (2) holds. Here, we assume when $\psi$ converges, $\mathbb{E}[\psi(\mathbf{x}; \alpha)|\psi(\mathbf{x}; \alpha) > 0] \approx \mathbb{E}_{\mathbf{x} \in Pos}[\psi(\mathbf{x}; \alpha)|\psi(\mathbf{x}; \alpha) > 0]$.

## C DISCUSSION OF CONSTRAINTS IN PROXY PROBLEM

In this section we discuss the properties and functionalities of the three constraints in the proxy problem as well as constructing example to illustrate how they work.

**Proxy Problem**    Minimization of the optimization problem

$$\min_{\psi} \quad \mathcal{L}(\psi) = -\log \mathbb{E}\left[\psi(\mathbf{x}; \alpha)|Q(\mathbf{x}) \in \mathcal{S}_{\mathcal{P}}\right] + \log \mathbb{E}\left[\psi(\mathbf{x}; \beta)|Q(\mathbf{x}) \in \mathcal{S}_{\mathcal{N}}\right], \tag{27}$$

yields a near-optimal solution for solving problem 11, with the constraints of theorem 4.4 being satisfied.

**Properties of $\alpha$ and $\beta$.**    We first discuss the properties of hyperparameters $\alpha$ and $\beta$. As show in appendix B, $\alpha > \beta$ is necessary to keep the second-order derivative positive. If we remove this constraints by simply setting $\alpha = \beta = 1$, then we observe that Eq. 23 actually becomes

$$\left.\frac{\partial h'}{\partial v_i}\right|_{\boldsymbol{v}=\boldsymbol{v}_0} = \frac{f}{g}(\alpha - \beta) = 0, \tag{28}$$

and we have the second-order derivative

$$\left.\frac{d^2 h''}{dx^2}\right|_{x=0} = \frac{f}{g}(\alpha - \beta)(\frac{1}{v_1} + \frac{1}{v_2}) = 0. \tag{29}$$

This means that while training the model, it is possible that the model's derivatives w.r.t. multiple nonzero dimensions $w_i$ of some $\boldsymbol{w}$ become 0 and the model might falls into local minima where it doesn't capture any logic classifier. We now discuss the situations where these risks actually exist.

The first and most simple case is when $p = q$, i.e.,

$$\mathcal{L}(\psi) = -\log \mathbb{E}_{\mathbf{x} \sim p}\left[\psi(\mathbf{x}; \alpha)\right] + \log \mathbb{E}_{\mathbf{x} \sim p}\left[\psi(\mathbf{x}; \alpha)\right] = 0. \tag{30}$$

Apparently in this case, training the model provides nothing because $\mathcal{L}(\psi)$ is a fixed scalar irrelevant to $\psi$. We argue that this is not a big problem since it requires $p$ and $q$, corresponding to the distributions of positive and negative data instances, to be the same.

By extending the above case, we can construct a more general situation. Recall that in theorem B.3, we have

$$h(\boldsymbol{w}, 1, 1) = \frac{\sum_i A_i(1)w_i}{\sum_i B_i(1)w_i} \leq \max_i \left\{\frac{A_i(1)}{B_i(1)}\right\} = \frac{A_m(1)}{B_m(1)}, \tag{31}$$

where we let $m$ be the dimension corresponding to the global maxima. If there are some dimension $k$ of $\boldsymbol{w}$ such that $A_k(1) = B_k(1) = 0$, then for any $\boldsymbol{w}'$ satisfying $w'_m > 0$ and $w'_k = 1 - w'_m$, we have $h(\boldsymbol{w}', 1, 1) = \frac{A_m(1)}{B_m(1)} = \sup\{h(\boldsymbol{w}, 1, 1)\}$. To solve this problem, we can add a normalization term to the original proxy problem, i.e.

$$\begin{aligned}
\mathcal{L}_{\text{Aug}}(\psi) &= \mathcal{L}(\psi) + \mathcal{L}_{\text{Norm}}(\psi) \\
&= -\log \mathbb{E}_{\mathbf{x} \sim p}\left[\psi(\mathbf{x}; 1)\right] + \log \mathbb{E}_{\mathbf{x} \sim q}\left[\psi(\mathbf{x}; 1)\right] - \lambda \log \mathbb{E}_{\mathbf{x} \sim p}\left[\psi(\mathbf{x}; 1)\right].
\end{aligned} \tag{32}$$

Thus, even when $A_k(1) = B_k(1) = 0$, the normalization term still encourages the model to assign larger value to $w_m$.

Another problematic situation still happens when there exists $i \neq j$ such that $A_i(1) = A_j(1) > 0$ and $B_i(1) = B_j(1) > 0$. This is not usual because it requires there exists two different logic classifiers $\varphi_1$ and $\varphi_2$ to have $\mathbb{E}_{\mathbf{x} \sim p}[\varphi_1(\mathbf{x})] = \mathbb{E}_{\mathbf{x} \sim p}[\varphi_2(\mathbf{x})]$ and $\mathbb{E}_{\mathbf{x} \sim q}[\varphi_1(\mathbf{x})] = \mathbb{E}_{\mathbf{x} \sim q}[\varphi_2(\mathbf{x})]$. Even when it happens, it's easy to deal with the problem, as during training we always reweight or sample different batches of training data, yielding a different data distribution $p'$ and $q'$, We can also set $\alpha = \beta > 1$, for example, $\alpha = \beta = 2$, to avoid such situations from happening.

**Tree structured paths constraints.**    We now discuss the second constraint in the problem. If we remove this constraints and there are circles for some unfixed nodes, we observe that $\mathcal{L}(\psi)$ w.r.t. the corresponding $\boldsymbol{w}$ might no more holds the form of Eq. 19. Note that in Eq. 7 we define the conjunction as $\mathcal{F}_i(\mathbf{x}) = \mathcal{F}_j(\mathbf{x})\mathcal{F}_k(\mathbf{x})$. If the evaluation of $\mathcal{F}_j$ and $\mathcal{F}_k$ both contains some $\boldsymbol{w}$, i.e. $\mathcal{F}_j = \boldsymbol{a}_j^T(\boldsymbol{w})^\alpha$ and $\mathcal{F}_k = \boldsymbol{a}_k^T(\boldsymbol{w})^\alpha$ where $\boldsymbol{a}_j$ and $\boldsymbol{a}_k$ are irrelevant quantities with $\boldsymbol{w}$, then $\mathcal{F}_j = \boldsymbol{a}_j^T(\boldsymbol{w}\boldsymbol{w}^T)^\alpha \boldsymbol{a}_k$ which no more holds the form of Eq. 19. Instead, we have

$$h(\boldsymbol{w}, \alpha, \beta) = \frac{\sum_i A_i(\alpha) \prod_j w_j^{n_j \alpha}}{\sum_i A_i(\alpha) \prod_j w_j^{n_j \alpha}}. \tag{33}$$

This form of $h(\boldsymbol{w}, \alpha, \beta)$ no longer keeps the good properties of Eq. 19, as it's second order derivatives w.r.t. some dimension of $\boldsymbol{w}$ no longer guarantee to be larger or equal than $0$. We provide an example to illustrate this.

Consider we want to learn logic rules of $Q(\mathbf{x}) \leftarrow \varphi_1(\mathbf{x}) := P_1(\mathbf{x}) \wedge P_1(\mathbf{x})$ and $Q(\mathbf{x}) \leftarrow \varphi_2(\mathbf{x}) := P_2(\mathbf{x}) \wedge P_2(\mathbf{x})$. This expression is redundant because the two terms of the conjunction are the same. Suppose we the model we used here is $\psi_1(\mathbf{x}; \alpha) = \psi_2(\mathbf{x}; \alpha)\psi_2(\mathbf{x}; \alpha)$ where $\psi_2(\mathbf{x}; \alpha) = \sum_i w_i^\alpha P_i(\mathbf{x})$ is a soft selection over all possible predicates. Unluckily, in the training data the logic rule $Q(\mathbf{x}) \leftarrow \varphi_1(\mathbf{x})$ and $Q(\mathbf{x}) \leftarrow \varphi_2(\mathbf{x})$ are never satisfied, i.e., $p(Q(\mathbf{x})|\varphi_1(\mathbf{x})) = p(Q(\mathbf{x})|\varphi_2(\mathbf{x})) = 0$, but for the rule $Q(\mathbf{x}) \leftarrow \varphi_3(\mathbf{x}) := P_1(\mathbf{x}) \wedge P_2(\mathbf{x})$ we have $p(Q(\mathbf{x})|\varphi_3(\mathbf{x})) > 0$. Then, it's easy to observe that the global minima of the proxy problem happens when $w_1 = w_2 = 0.5$.

Although this situation disobeys with the conclusions of the proxy problem, both the training data, the target logic rules and construction of LPs are ill-conditioned, as we placed the same $\psi_2$ between the conjunction as well as in the training data we are unable to find any high-precision logic rules. We argue that in reality this can often be avoided because we can increase the expressiveness of the model, and set $\alpha$ and $\beta$ to a relative large value so that even the above situation happens, because $w_1^\alpha = w_2^\alpha$ are rather small values, it provides little stimulation to the model, and thus the model is encouraged to choose other rule structures that better explains the data. Also, because during training when the model is not at convergence, dimensions of $\boldsymbol{w}$ are rather small values, and the higher order terms $\prod_j w_j^{n_j \alpha}$ where $\sum_j n_j$ is large are much smaller than oridinary terms, which means they provide little influence to the overall derivatives of $\boldsymbol{w}$, so this problem is less serious.

**Negations.** When training completes and model converges, negations do not influence the validity of the model. However, during training, negations might be tricky to deal with, as adding negations to the training LPs might lead to the coefficients of corresponding $w_i$ in $h(\mathbf{x}, \alpha, \beta)$ being negative. The following example illustrated how negations might influence the training process.

Consider we want to learn the logic rules $Q(\mathbf{x}) \leftarrow \varphi_1(\mathbf{x}) := \neg P_1(\mathbf{x})$ and $Q(\mathbf{x}) \leftarrow \varphi_2(\mathbf{x}) := \neg P_2(\mathbf{x})$, and we construct a model of $\psi_1(\mathbf{x}; \alpha) = 1 - \psi_2(\mathbf{x}; \alpha)$ where $\psi_2(\mathbf{x}; \alpha) = \sum_i w_i^\alpha P_i(\mathbf{x})$. Suppose $\mathbb{E}[\varphi_1(\mathbf{x})|Q(\mathbf{x}) = 1] = \mathbb{E}[\varphi_2(\mathbf{x})|Q(\mathbf{x}) = 1] = E_1$, $\mathbb{E}[\varphi_1(\mathbf{x})|Q(\mathbf{x}) \neq 1] = \mathbb{E}[\varphi_2(\mathbf{x})|Q(\mathbf{x}) \neq 1] = E_2$ Then, by letting $\psi_2(\mathbf{x}; \alpha) = 0.5^\alpha P_1(\mathbf{x}) + 0.5^\alpha P_2(\mathbf{x})$, we observe that

$$
\begin{aligned}
\mathcal{L}(\psi) &= -\log \mathbb{E}\left[\psi(\mathbf{x}; \alpha)|Q(\mathbf{x}) = 1\right] + \log \mathbb{E}\left[\psi(\mathbf{x}; \beta)|Q(\mathbf{x}) \neq 1\right] \\
&= -\log\left(0.5^\alpha \left(\mathbb{E}[\varphi_1(\mathbf{x})|Q(\mathbf{x}) = 1] + \mathbb{E}[\varphi_2(\mathbf{x})|Q(\mathbf{x}) = 1]\right)\right) \\
&\quad + \log\left(0.5^\beta \left(\mathbb{E}[\varphi_1(\mathbf{x})|Q(\mathbf{x}) \neq 1] + \mathbb{E}[\varphi_2(\mathbf{x})|Q(\mathbf{x}) \neq 1]\right)\right) \\
&= -\alpha + \beta + \text{constant}.
\end{aligned}
\tag{34}
$$

Thus, we can see $\mathcal{L}(\psi)$ is monotonic decreasing w.r.t. $\alpha - \beta$, and such $\psi_2$ can reach a smaller value than $\varphi_1$ and $\varphi_2$ once $\alpha - \beta$ is sufficiently large. We can set $\alpha = \beta$ to a relative small value and carefully select the target LPs when trying to assign negations to them to avoid the problem from happening.

So far we have discussed the situations when the model might fail to converge if we discard the constraints of the proxy problem. As we can see, most of these invalidate situations can be avoided by setting $\alpha = \beta = 1$ or $2$ as well as providing a reasonable grammar of target logic rules. We argue that even if sometimes the model fails to converge at capturing a logic classifier, we can reinitialize the parameters of the relevant LPs randomly and train the model again so that it converges at other minimum points.

## D   EXPERIMENT DETAILS

In this section we explain the detailed model configuration for each experiment.

### D.1   ILP TASKS

The 20 ILP tasks introduced by Evans & Grefenstette (2018) cover problems from integer recognition, family tree reasoning, general graph algorithms and so on. We briefly summarize them here.

**Task 1-6** In task 1 to task 6 we are provided with natural numbers from $0$ to $9$, which are defined as follows:

$$\mathcal{S}_\mathcal{B} = \{zero(0), succ(0, 1), ..., succ(8, 9)\}. \tag{35}$$

The target is to learn to recognize predecessor, even / odd numbers, the less-than relation and divisible by $3$ or $5$.

**Task 7-8** Task 7-8 requires us to learn the relation member and length of a list. Nodes in a list is encoded as follows: $cons(x, y)$ if the node after $x$ is $y$, and $value(x, y)$ if the value of node $x$ is $y$. Two background statements are given, corresponding to the list $[4, 3, 2, 1]$ and $[2, 3, 2, 4]$.

**Task 9-14** In task 9-14 we are provided with different facts about family relations, and we need to learn the relations including son, grandparent, husband, uncle, relatedness and father. An example of rules would be $son(x, y) \leftarrow father(x, y) \wedge \varphi(x), \varphi(x) := brother(x, y) \vee father(x, y)$ where $\varphi$ implies the $male$ property.

**Task 15-20** In task 15-20 we are provided with labeled directed graphs, and we are asked to learn general concepts of graph algorithms. These tasks includes to learn whether a node is adjacent to a red node; whether a node has at least two children; whether a graph is well-colored, i.e., to identify if there are two adjacent nodes of the same color; whether two nodes are connected; and recognize graph cycles.

As a principled approach, $\partial$ILP (Evans & Grefenstette (2018)) is able to solve all these tasks. However, they need to construct different language templates and program templates for each task, for example, to solve the even numbers problem, they use the following templates:

$$
\begin{aligned}
P_e &: \{zero/1, succ/2\} \\
P_i &: \{target/1, pred/2\} \\
\tau_{target,1} &= (h = target, n_\exists = 0, int = False) \\
\tau_{target,2} &= (h = target, n_\exists = 1, int = True) \\
\tau_{pred,1} &= (h = pred, n_\exists = 1, int = False) \\
\tau_{pred,2} &= \text{null}.
\end{aligned} \tag{36}
$$

In contrast, we use a unified model to solve the tasks. Our grammar of logic classifiers are as follows:

$$
\begin{aligned}
\varphi(x) &:= P(x) \mid \varphi(x, x) \mid \exists y : \varphi(y) \wedge \varphi(x, y), \\
\varphi(x, y) &:= P(x, y) \mid \varphi(x) \wedge \varphi(y) \wedge \varphi(x, y) \mid \exists z : \varphi(x, z) \wedge \varphi(z) \wedge \varphi(z, y).
\end{aligned} \tag{37}
$$

For some tasks if there are no observed predicates of arity 1 or 2, we simply create an invented predicates with all 1 values for every variables **x**. On all tasks we set $\alpha = \beta = 1$. The construction of inference network of the model and how we train the model is the same as described in Sec. 4.1, Sec. 4.2 and Sec. 4.4. The correctness of learnt model is confirmed by checking whether all positive queries are predicted as true and all negative queries are predicted as false by the model.

### D.2 SYSTEMATICITY TESTS

All model configurations are the same as in the ILP tasks.

### D.3 KNOWLEDGE GRAPH COMPLETION

The statistics of datasets are summarised as follows.

Table 8: Statistics of datasets.

| Dataset | #Entities | #Relations | #Trains | #Validation | #Test |
|---------|-----------|------------|---------|-------------|-------|
| FB15k-237 | 14514 | 237 | 272115 | 17535 | 20466 |
| WN18RR | 40943 | 11 | 86835 | 3034 | 3134 |
| Kinship | 104 | 25 | 3206 | 2137 | 5343 |
| UMLS | 135 | 46 | 1959 | 1306 | 3264 |

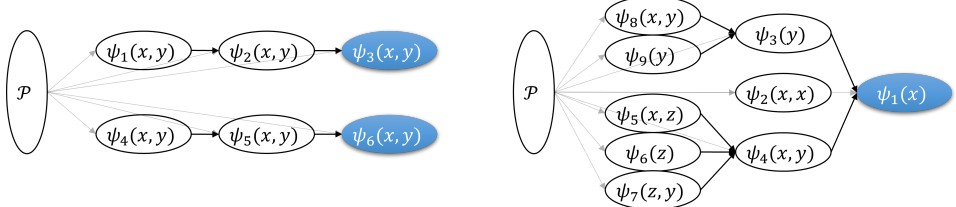

Figure 3: Illustration of two inference networks generated by procedure in Sec. 4.3. We remove $\mathcal{F}_i$ nodes for notation clarity. Output nodes are colored in blue. The left figure corresponds to $\varphi(x,y) := A(x,y) \mid \exists z : \varphi(x,z) \wedge A(z,y)$, which is applicable to the systematicity tests and knowledge graph completion tasks. The right figure corresponds to $\varphi(x) := A(x) \mid \exists y : \varphi(x,y) \wedge \varphi(y)$ and $\varphi(x,y) := A(x,y) \mid \exists z : \varphi(x,z) \wedge \varphi(z) \wedge \varphi(z,y)$, which is used in the ILP tasks.

Generally, knowledge graphs are much noisier than ILP tasks, as shown in Fig. 2, most learnt rules have a rather small value of accuracy compared to ILP tasks (correct rule accuracy is 1) and systematicity tests (learnt rule accuracy usually more than $0.8$). To handle such uncertainty, we set the number of inference iterations to be 1 and learn more rules for each predicate. On all tasks our grammar is the same as in ILP tasks. Because there are no unary predicates in KG, the resulted grammar is essentially

$$\varphi(x,y) := \exists z : \varphi(x,z) \wedge \varphi(z,y), \tag{38}$$

which is corresponds to chain-like rules. On all KG completion tasks we learn rules of length 3. We create negative statements via negative sampling. Instead of removing the proved queries, we reduce the weights of corresponding queries with the fixed ratio $0.8$. We train for 400 times for each relation and remove duplicated rules. When testing, we choose different update functions for inference, including the original ones in Eq. 9, a modification of the original ones where we set the restriction $\psi(\mathbf{x}) = \max\{\psi(\mathbf{x}), 1\}$ thus the evaluation value provided by $\psi$ is exactly the same as the corresponding logic classifier, and a multi-layer perceptron based on the validation data.

For evaluation, we use the standard filtered ranking (Bordes et al. (2013)) metrics, including Mean Rank (MR), Mean Reciprocal Rank (MRR) and Hit@k (H@k). When there are multiple tail nodes assigned with the same score, we compute the expectation of each evaluation metric over all random shuffles of entities (Qu et al. (2021)).

After all, in all of the experiments, we use Adam (Kingma & Ba (2015)) optimizer with $lr = \{0.01, 0.1\}$. We let $\alpha = \beta = \{1, 2\}$. We generate the structural parameters of LPs $\boldsymbol{w}$ using a softmax function as follows:

$$\boldsymbol{w} = \text{softmax}(\boldsymbol{w}'), \tag{39}$$

where $\boldsymbol{w}' \in \mathbb{R}^{n_w}$ are real-valued vectors with no constraints. We prepossess the data to add permutations for ordinary predicates, for example, for every 2-ary statement $P(a, b)$ in data we create an invented one $P'(b, a)$. For randomly initialized parameters, we draw them independently from Gaussian distribution $\mathcal{N}(0, 1)$, but this is not necessary and other distributions (uniform, Xavier, ...) also work well. Illustrations of inference network architecture constructed with the procedure discribed in Sec. 4.3 are in Fig. 3.

## E    DISCUSSION OF TIME COMPLEXITY

In this section we discuss the model complexity. Suppose the inference network is composed of $N$ LPs. Suppose the arities of predicates and LPs are at most $n$. We now discuss the time complexity of each part of the inference model.

**General time complexity**    We first take a look at a single LP's behaviour, where we assume that all other LPs in the network are already evaluated. From the definitions of LPs Eq. 6 and Eq. 7, we can see that evaluating one LP invloves:

$$\psi(\mathbf{x}; \alpha) = (\boldsymbol{w}^T)^\alpha [\mathcal{F}_1(\mathbf{x}; \alpha), \mathcal{F}_2(\mathbf{x}; \alpha), \mathcal{F}_3(\mathbf{x}; \alpha), ...]^T. \tag{40}$$

Since the total amount of $\mathcal{F}_i$ is limited and does not scale as the network or input data grows, this step costs

$$T(\psi) \leq \sum_i T(\mathcal{F}_i) + O(|\mathcal{V}|^n), \tag{41}$$

where we let $T(\psi)$ be the time complexity for evaluating $\psi(\mathbf{x}; \alpha)$ for all $\mathbf{x}$, $T(\mathcal{F}_i)$ be the time complexity for evaluating $\mathcal{F}_i(\mathbf{x}; \alpha)$ for all $\mathbf{x}$, etc. For each $\mathcal{F}_i$, we have:

$$
\begin{aligned}
\mathcal{F}_i(\mathbf{x}; \alpha) &= \left[ P_1(\mathbf{x}), P_2(\mathbf{x}), ..., P_{|\mathcal{P}|}(\mathbf{x}), \psi_i'(\mathbf{x}) \right] \left( \boldsymbol{w}^{(i)} \right)^\alpha &\Longleftrightarrow T(\mathcal{F}_i) &\leq O(|\mathcal{P}||\mathcal{V}|^n), \\
\mathcal{F}_i(\mathbf{x}; \alpha) &= \mathcal{F}_j(\mathbf{x}; \alpha) \mathcal{F}_k(\mathbf{x}; \alpha) &\Longleftrightarrow T(\mathcal{F}_i) &\leq T(\mathcal{F}_j) + T(\mathcal{F}_k) + O(|\mathcal{V}|^n), \\
\mathcal{F}_i(\mathbf{x}; \alpha) &= [\mathcal{F}_j(\mathbf{x}; \alpha), \mathcal{F}_k(\mathbf{x}; \alpha)] \left( \boldsymbol{w}^{(i)} \right)^\alpha &\Longleftrightarrow T(\mathcal{F}_i) &\leq T(\mathcal{F}_j) + T(\mathcal{F}_k) + O(|\mathcal{V}|^n), \\
\mathcal{F}_i(\mathbf{x}; \alpha) &= 1 - \mathcal{F}_j(\mathbf{x}; \alpha) &\Longleftrightarrow T(\mathcal{F}_i) &\leq T(\mathcal{F}_j) + O(|\mathcal{V}|^n), \\
\mathcal{F}_i(\mathbf{x}; \alpha) &= \sum_{\mathbf{y}} \mathcal{F}_j(\mathbf{x}, \mathbf{y}; \alpha) &\Longleftrightarrow T(\mathcal{F}_i) &\leq T(\mathcal{F}_j) + O(|\mathcal{V}|^n), \\
\mathcal{F}_i(\mathbf{x}, \mathbf{y}; \alpha) &= \mathcal{F}_j(\mathbf{x}; \alpha) &\Longleftrightarrow T(\mathcal{F}_i) &\leq T(\mathcal{F}_j) + O(|\mathcal{V}|^n).
\end{aligned}
\tag{42}
$$

Thus we can see that for one $\psi$, we have

$$T(\psi) \leq \sum_{\mathcal{F}_i} O(|\mathcal{P}||\mathcal{V}|^n) = O(|\mathcal{P}||\mathcal{V}|^n). \tag{43}$$

To evaluate all nodes in the network, we simple evaluate LPs one by one in topological order, which directly gives a total time complexity of

$$T(\Psi) = O(|\mathcal{P}||\mathcal{V}|^n N). \tag{44}$$

The update procedure for one $\mathbf{x}$ is a function with $R_\Psi$ (amount of learnt rules) inputs, and all implementations we introduced here all make the evaluation of the function $O(R_\Psi)$), so the evaluation over all $\mathbf{x}$ takes $O(|\mathcal{V}|^n R_\Psi)$ time.

**Time complexity on sparse graphs** In reality often the input data is sparse, and the $|\mathcal{V}|^n$ term in time complexities can be reduced significantly. Here, we analyse the time complexity of the model under knowledge graph completion experiments.

Suppose the input graph has $|\mathcal{V}|$ nodes, $|\mathcal{P}|$ predicates and $M$ edges, and we are learning chain-like rules of length $L$. Thus, each output unit, corresponding to a rule, is composed of $L$ LPs, and can be written as follows:

$$\psi(x, y; \alpha) = \sum_{z_1, z_2, ..., z_{L-1}} \mathcal{P}_1(x, z_1; \alpha) \mathcal{P}_2(z_1, z_2; \alpha) ... \mathcal{P}_L(z_{L-1}, y; \alpha), \tag{45}$$

where

$$\mathcal{P}_i(x, y; \alpha) = [P_1(x, y), P_2(x, y), ...] \boldsymbol{w}_i^\alpha. \tag{46}$$

We can efficiently calculate $\psi(x, y; \alpha)$ for all nodes $y$ in the knowledge graph with the same source $x$. The algorithm is an extension of $L$-step breadth first search described as follows. $S_L$ maps nodes $y$ with nonzero value $\psi(x, y; \alpha)$ to $\psi(x, y; \alpha)$.

---

**Algorithm 1** Inference in sparse graphs

---

**Input:** graph $\mathcal{G}$, predicates $\mathcal{P}$, source node $x$, rule length $L$, model parameters $\alpha, \boldsymbol{w}_1, \boldsymbol{w}_2, ..., \boldsymbol{w}_L$.
**Output:** $S_L$.
 1: **function** EVALUATE($x, \mathcal{G}, L, \alpha, \boldsymbol{w}_1, \boldsymbol{w}_2, ..., \boldsymbol{w}_L$)
 2:     $S_0 \leftarrow$ MAP($\emptyset$)
 3:     $S_0[x] \leftarrow 1.0$
 4:     **for** $l \leftarrow 1$ to $L$ **do**
 5:         $S_l \leftarrow$ MAP($\emptyset$)
 6:         **for** $P \in \mathcal{P}, s \in S_{l-1}, t \in \mathcal{N}_P(s)$ **do**
 7:             **if** $t \notin S_l$ **then**
 8:                 $S_l[t] \leftarrow 0.0$
 9:             **end if**
10:             $S_l[t] \leftarrow S_l[t] + (\boldsymbol{w}_l[P])^\alpha \ S_{l-1}[s]$
11:         **end for**
12:     **end for**
13:     **return** $S_L$
14: **end function**

---

Here, we denote $\mathcal{N}(s)$ as the neighbors of $s$ in the KG, and $\mathcal{N}_P(s)$ as the neighbors of $s$ connected by edge type $P$. Thus, one run of the function takes at least

$$
\begin{aligned}
T_{\text{single}} &= \sum_l \sum_{s \in S_{l-1}} \sum_{P \in \mathcal{P}} \sum_{t \in \mathcal{N}_P(s)} O(1) \\
&= \sum_{l=1}^{L} \sum_{s \in S_{l-1}} O(|\mathcal{N}(s)|)
\end{aligned}
\tag{47}
$$

Denoting $N^{(k)}$ as the max amount of nodes in a node's $k$-hop subgraph, we have

$$
\begin{aligned}
T_{\text{single}} &\leq \sum_{l=1}^{L} N^{(l-1)} O(N^{(1)}) \\
&\leq O(L N^{(L)} N^{(1)}).
\end{aligned}
\tag{48}
$$

Thus, one-time of evaluating a total number of $R_\Psi$ rules for all node pairs in the graph takes

$$
\begin{aligned}
T &\leq R_\Psi |\mathcal{V}| T_{\text{single}} \\
&= O(R_\Psi L N^{(L)} N^{(1)} |\mathcal{V}|).
\end{aligned}
\tag{49}
$$

Since in sparse graphs we often have $N^{(1)} << N^{(L)} << |\mathcal{V}|$, this estimation of time complexity (Eq. 49) is much less than the original one (Eq. 44), which is $O(R_\Psi L |\mathcal{P}| |\mathcal{V}|^3)$ in this case.

## F    CASE STUDIES

In this section we illustrate part of the logic rules we learned on ILP tasks, systematicity tests and knowledge graph completion.

Table 9: Sparsity of LP-layer.

| Task | Converge Ratio |
|------|----------------|
| ILP $\alpha = 1$ | 0.915 |
| Systematicity $\alpha = 1$ | 0.813 |
| Systematicity $\alpha = 2$ | 0.956 |
| KG completion $\alpha = 1$ | 1.0 |

| Target | $\leftarrow$ | Rules |
|--------|--------------|-------|
| DivisibleBy3$(x)$ | $\leftarrow$ | $\psi_1(x) = \text{Zero}(x)$ |
| | $\leftarrow$ | $\psi_2(x) = \exists y : \psi_3(y) \wedge \text{Succ}(y, x)$ |
| | | $\psi_3(x) = \exists y : \psi_4(y) \wedge \text{Succ}(y, x)$ |
| | | $\psi_4(x) = \exists y : \text{DivisibleBy3}(y) \wedge \text{Succ}(y, x)$ |
| AdjacentToRed$(x)$ | $\leftarrow$ | $\psi_1(x) = \exists y : \text{Red}(y) \wedge \text{Edge}(x, y)$ |
| Connect(x, y) | $\leftarrow$ | $\psi_1(x, y) = \text{Edge}(x, y)$ |
| | $\leftarrow$ | $\psi_2(x, y) = \exists z : \text{Connect}(x, z) \wedge \text{Edge}(z, y)$ |
| Grandmother$(x, y)$ | $\leftarrow$ | $\psi_1(x, y) = \exists z : \text{Brother}(x, z) \wedge \text{Grandmother}(z, y)$ |
| | $\leftarrow$ | $\psi_2(x, y) = \exists z : \text{Father}(x, z) \wedge \text{Mother}(z, y)$ |
| Causes$(x, y)$ | $\leftarrow$ | $\psi_1(x, y) = \exists z_1, z_2 : \text{Contains}(z_1, x) \wedge \text{LocationOf}(z_1, z_2)$ $\wedge \text{OccursIn}(z_2, y)$ |
| | $\leftarrow$ | $\psi_2(x, y) = \exists z_1, z_2 : \text{Contains}(z_1, x) \wedge \text{LocationOf}(z_1, z_2)$ $\wedge \text{Complicates}(y, z_2)$ |
| | $\leftarrow$ | $\psi_3(x, y) = \exists z_1, z_2 : \text{IngredientOf}(z_1, x) \wedge \text{IsA}(z_1, z_2)$ $\wedge \text{Causes}(z_2, y)$ |
| | $\leftarrow$ | $\psi_4(x, y) = \exists z_1, z_2 : \text{IngredientOf}(z_1, x) \wedge \text{InteractsWith}(z_2, z_1)$ $\wedge \text{Causes}(z_2, y)$ |

## G    ADDITIONAL ABLATION STUDY

In this section we provide empirical results for additional ablation study.

### G.1    SPARSITY OF LP-LAYER

We run LP-layer on the experiments used in this paper and obtain the sparisity of learnt parameters. For each $w$ used for prediction, if after training $\max\{w_1, w_2, ...\} \geq 0.99$ we simply regard $w$ as being converged. Then, we obtain the results in Tab. 9.

### G.2    DEPTH OF NETWORKS

We run LP-layer and LP-tree with different depths $\{3, 5, 10\}$ on ILP and Systematicity tests. Depth of 3 cannot complete some of the ILP tasks which requires more reasoning steps. Depth of 5 and 10 are both able to complete ILP tasks. Results on Systematicity tests are very similar with depth $3, 5, 10$, as is shown in Tab. 10.

### G.3    CLIPPING ON LP

We now study the effects of whether to restrict the range of $\psi$ to be $[0, 1]$ by simply setting $\psi(\mathbf{x}; \alpha) \leftarrow \min\{\psi(\mathbf{x}; \alpha), 1\}$. The results are shown in Tab. 11.

Table 10: Depth of networks.

|  | 4Hops | 5Hops | 6Hops | 7Hops | 8Hops | 9Hops | 10Hops |
|---|---|---|---|---|---|---|---|
| DLP-tree-3 | .990 | .994 | 1.0 | .995 | .997 | .996 | 1.0 |
| DLP-layer-3 | .991 | .993 | 1.0 | .995 | .997 | .996 | 1.0 |
| DLP-tree-5 | .995 | .994 | 1.0 | .997 | .997 | .996 | 1.0 |
| DLP-layer-5 | .995 | .994 | 1.0 | .997 | .997 | .996 | 1.0 |
| DLP-tree-10 | .995 | .994 | 1.0 | .993 | .997 | .996 | 1.0 |
| DLP-layer-10 | .995 | .994 | 1.0 | .997 | .997 | .996 | 1.0 |

Table 11: Setting $\psi(\mathbf{x}; \alpha) \leftarrow \min\{\psi(\mathbf{x}; \alpha), 1\}$.

| Method | Kinship | | | | | UMLS | | | | |
|---|---|---|---|---|---|---|---|---|---|---|
| | MR | MRR | H@1 | H@3 | H@10 | MR | MRR | H@1 | H@3 | H@10 |
| With clipping | 3.7 | 0.639 | 0.498 | 0.725 | 0.926 | 3.2 | 0.800 | 0.689 | 0.892 | 0.957 |
| Without clipping | 3.7 | 0.645 | 0.504 | 0.733 | 0.927 | 3.1 | 0.810 | 0.708 | 0.896 | 0.959 |

