# OpenReview forum: "Differentiable Logic Programming for Probabilistic Reasoning"
_ICLR.cc/2023/Conference — Submitted to ICLR 2023_

### Official Review · Reviewer_mi4W · 2022-10-25

**Confidence:** 2
**Correctness:** 3
**Technical Novelty And Significance:** 3
**Empirical Novelty And Significance:** 3
**Recommendation:** 6

**Clarity, Quality, Novelty And Reproducibility:**

Clarity,  Mostly clear
Quality, Loks good, especially in reviewing related work.
Novelty I would like a more convincing argument of that
Reproducibkitty :ok

**Strength And Weaknesses:**

This paper presents a theory for obtaining derivatves of logic programs Several researches worked on this, Domingo s and colleagues had a relatively simple model for mlns, Problog used Derivatives from BDDS, Last we have graph Rule construction: iit is kind of reminiscent of Tensorlog

Results look good (including variance).

Isues: What is novel here?

Are you constructing a probabilistic system or just a NN?
Please explain more in detail how this work evolves over previous packages/

**Summary Of The Paper:**

This paper presents a  derivcate construction mechanims for logical formulae. Results are very good.

**Summary Of The Review:**


Maybe my fault: thereś a lot of good suff here, but it didn;t entirely convince me, esoecially when comparing  to related work.

---

> ### Author Response · Authors · 2022-11-17
> **Author Response**
>
> We thank the reviewer for the insightful comments! We have updated the paper to improve the clarity and readability. The reviewer is welcomed to check the revised paper.
>
> > What is novel here? Please explain more in detail how this work evolves over previous packages.
>
> Our work is related to previous rule-learning methods. From the first glance the techniques used in our work is similar to differentiable rule learning methods including NeuralLP, DRUM and their extensions. Compared to these works, the contributions we made are:
> 1. We learn sparse solutions, where each local optimal of the continuous proxy problems explicitly reveals a symbolic rule structure. The previous series of work (including NeuralLP, DRUM, NTP, CTP and so on) do not actually learn logic rules (this is also addressed in Sec. 6 of the survey [R1]). The basic ideas of NeuralLP, DRUM and their extensions are to approximate the simultaneously evaluation of all possible logic rules in exponential space, while the ideas of NTP, CTP and their extensions are to use neural networks to simulate backward chaining. Also, dense vector representations for each entity are used in NTP, CTP, etc. Compared to these methods, our method is able to generate sparse solutions by optimizing on the continuous proxy problem where each local optimal of the continuous proxy problems explicitly reveals a symbolic rule structure. The experiments show the effectiveness of our sparse solutions, compared with the dense vector representations of NeuralLP, DRUM, NTP, etc.
> 2. We are able to learn more complex rules. NeuralLP, DRUM are only able to learn chain-like rules on KG with only binary predicates. NTP, CTP, etc. are also only evaluated on small dataset where only binary predicates are used. Moreover, none of these differentiable learning methods is able to reach comparable results in the experiments.
>
> As discussed above, if we focus on "the results produced by the model" rather than "the techniques used in the model", our work is essentially more like explicit rule mining methods such as AMIE, RNNLogic, etc. because they both learn logic rules explicitly. However, traditional rule learning methods still requires to perform combinatory search to find useful rules. If we want to learn chain-like rules (which is common for these methods) of length $L$, the search space of these methods grows exponential w.r.t. $L$, while in our approach the computational complexity grows linearly w.r.t. $L$.
>
> Recently, some methods including reinforcement learning, RNN, etc. have been used to enhance the rule searching procedure. For example, RNNLogic uses a RNN learn to generate useful rules, which reduces the search space of logic rules. However, the participation of RL / RNN also largely restrict the expressiveness of learnt rules. For example, RNN is only able to generate chain-like rules with binary predicates. Compared with them, our model is able to learn more complex rules.
>
> > Are you constructing a probabilistic system or just a NN?
>
> We are constructing a probabilistic system, which is able to (1) mine logic rules and (2) learn to perform probabilistic reasoning based on the rules.
>
> > it didn't entirely convince me, especially when comparing to related work.
>
> We have discussed the novelty of our approach above. Besides, in the experiments, we have compared our method with different baselines on different tasks. The goal is to provide a general and flexible rule-learning framework that is able to solve a variety of tasks. For example, it's hard to extend RNNLogic to be able to solve the ILP and CLUTRR tasks. Instead, by using the same grammar and structures of LPs, our model is able to achieve results comparable to (even better than) sota (R5 for CLUTRR, RNNLogic+ for KG completion) on all these tasks.
>
> Compared with other differentiable learning methods, our rule learning framework is analogous to boosting methods, where each single logic rule is analogous to a weak classifier. Each logic rule is learnt by maximizing its accuracy to better capture the local patterns of data, while the whole learning procedure (including reweighting of data) makes it possible to learn the global patterns of data. The effect of learning framework of RNNLogic is somehow alike to ours, but their EM-based algorithm is tricky to deal with.
>
> The two approaches R5 and RNNLogic are comparable to our methods. We have tried to extend RNNLogic to solve the CLUTRR task. The results are as follows. As for R5, the authors specially use a section in their paper to discuss why R5 is unsuitable for KG completion tasks, as R5 targets at finding the correct relation given the head and tail, i.e. $(h,?,t)$, thus as the authors of R5 says, "*It is also non-trivial to extend our research to tackle KBC tasks*". More reasons about why R5 is infeasible for KBC problems can be seen at the paragraph 3 in Appendix E of the paper [R2].

---

> > ### Author Response · Authors · 2022-11-17
> > **Author Respond 2**
> >
> > We have tried to modify RNNLogic to make it work on the relation prediction task CLUTRR and chose the best results. The results are as follows. There are total $40$ relations in the dataset, so randomly choosing provides $0.025$ accuracy.
> >
> > |           |  4-Hop   | 5-Hop  | 6-Hop | 7-Hop | 8-Hop | 9-Hop | 10-Hop
> > |  ----     |  ----  | ----  | ------ | ------- | ------- | ------- | ------- |
> > |   RNNLogic     | 0.091  | 0.066 | 0.071 | 0.061 | 0.054 | 0.043 | 0.046
> >
> >
> > [R1] Luc De Raedt, Sebastijan Dumancic, Robin Manhaeve, Giuseppe Marra: From Statistical Relational to Neuro-Symbolic Artificial Intelligence. IJCAI 2020: 4943-4950.
> >
> > [R2] Lu, Shengyao et al. “R5: Rule Discovery with Reinforced and Recurrent Relational Reasoning.” ICLR 2022.

---

### Official Review · Reviewer_kdMZ · 2022-10-25

**Confidence:** 4
**Correctness:** 3
**Technical Novelty And Significance:** 2
**Empirical Novelty And Significance:** Not applicable
**Recommendation:** 3

**Clarity, Quality, Novelty And Reproducibility:**

The clarity of this paper needs significant improvement, especially regarding notations and their meanings. The ideas presented in this paper have been explored by many recent papers, thus the novelty is little. There are extensive comparisons with many baselines, which is great, however, the empirical improvement is not significant at all.

**Strength And Weaknesses:**

Strengths:
- this paper uses many recent baselines and three different kinds of datasets in the empirical evaluations
- many performance ablation studies have been presented

Weaknesses:
- the writing is not easy to follow, particularly, heavy use of (undefined/unexplained) notations makes the reading quite challenging. For instance, proposition 4.2 is especially hard to understand.
- the idea of relaxing logical operations as numerical operations and attaching logical rules with weights is well-known
- using differentiable learning to discover useful logical rules is also not new (see the survey paper R1)
- the so-called generative definition (Equation 1) is essentially a simple template, which may be more generally viewed as grammar. Then the construction of networks (or rule generation process) is a simple instance of the well-known Syntax-guided synthesis (SyGuS) in the programming language community (see R2).
- the empirical evaluation shows very little improvement. Particularly, around 1% or less improvement on three knowledge graph completion datasets (i.e., Kinship, FB15k-237, WN18RR).

[R1] Luc De Raedt, Sebastijan Dumancic, Robin Manhaeve, Giuseppe Marra: From Statistical Relational to Neuro-Symbolic Artificial Intelligence. IJCAI 2020: 4943-4950

[R2] 	Rajeev Alur, Rastislav Bodík, Eric Dallal, Dana Fisman, Pranav Garg, Garvit Juniwal, Hadas Kress-Gazit, P. Madhusudan, Milo M. K. Martin, Mukund Raghothaman, Shambwaditya Saha, Sanjit A. Seshia, Rishabh Singh, Armando Solar-Lezama, Emina Torlak, Abhishek Udupa: Syntax-Guided Synthesis. Dependable Software Systems Engineering 2015: 1-25


**Summary Of The Paper:**

This paper proposes a differentiable logic programming (DLP) framework, which relaxes logical operations (e.g., AND, OR) in a rule with numerical operations (e.g., multiplication, addition). DLP can learn logical rules and weights. The key idea is to (recursively) generate logical rules by following a chain-like pattern (specified in Equation 1) with conjunction, disjunction, and negation. Candidate logical rules are associated with weights, which are learned through minimized a proxy loss. The empirical evaluation shows that DLP achieves similar or slightly better performance compared to a number of baselines.

**Summary Of The Review:**

This paper studies an important and interesting challenge in AI -- learning logical rules and weights. The approach of using numerical relaxation and simple template (or so-called generative definition in the paper) to facilitate rule learning is well-known. It's difficult to see the actual novelty besides an obscure way of formalization. And, the practical significance is unclear given the current evaluations.

---

> ### Author Response · Authors · 2022-11-17
> **Author Response 1**
>
> We thank the reviewer for the insightful comments! We have updated the paper to improve the clarity and readability. The reviewer is welcomed to check the revised paper.
>
> Before we proceed to address your questions, one thing we want to clarify first is that in this work, the proxy loss is not used to learn the weights of the rules. Instead, it is used to learn the symbolic structures of the rules (i.e., corresponds to the rule generation process). Also, in all experiments of this paper we use exactly the same network to learn different rules, so we do not need to search for different network architectures to learn them.
>
> > the writing is not easy to follow, particularly, heavy use of (undefined/unexplained) notations makes the reading quite challenging. For instance, proposition 4.2 is especially hard to understand.
>
> We have updated the paper to make it easier for reading, as well as simplifying the notations and making the expressions more accurate. We have also provided a list of updates. The Proposition 4.2 is stated to better introduce the general learning framework of our approach. In fact it's just like the well-known boosting methods: we first learn a logic rule based on the original data, and then we reweight the data and learn the rules on the data again. We have rewrite Proposition 4.2 to make it more clear.
>
> > the idea of relaxing logical operations as numerical operations and attaching logical rules with weights is well-known; using differentiable learning to discover useful logical rules is also not new (see the survey paper R1)
>
> We have thoroughly investigated the survey paper [R1]. The idea of relaxing logical operations as numerical operations, attaching weights, and using differentiable learning to discover useful logical rules is well-known, but our work makes two major contributions including:
>
> 1. We learn sparse solutions, where each local optimal of the continuous proxy problems explicitly reveals a symbolic rule structure. The previous series of work (including NeuralLP, DRUM, NTP, CTP and so on) do not actually learn logic rules (this is also addressed in Sec. 6 of the survey [R1]). The basic ideas of NeuralLP, DRUM and their extensions are to approximate the simultaneously evaluation of all possible logic rules in exponential space, while the ideas of NTP, CTP and their extensions are to use neural networks to simulate backward chaining. Also, dense vector representations for each entity are used in NTP, CTP, etc. Compared to these methods, our method is able to generate sparse solutions by optimizing on the continuous proxy problem where each local optimal of the continuous proxy problems explicitly reveals a symbolic rule structure. The experiments show the effectiveness of our sparse solutions, compared with the dense vector representations of NeuralLP, DRUM, NTP, etc.
> 2. We are able to learn more complex rules. NeuralLP, DRUM are only able to learn chain-like rules on KG with only binary predicates. NTP, CTP, etc. are also only evaluated on small dataset where only binary predicates are used. Moreover, none of these differentiable learning methods is able to reach comparable results in the experiments.
>
> > the so-called generative definition (Equation 1) is essentially a simple template, which may be more generally viewed as grammar. Then the construction of networks (or rule generation process) is a simple instance of the well-known Syntax-guided synthesis (SyGuS) in the programming language community (see R2).
>
> We have carefully read the SyGuS paper. A number of synthesis methods have been introduced in the paper, and our work differs from them in that we transform the original discrete search problem into a continuous optimization problem, i.e., we generate rules by minimizing the proxy loss. Our network architecture is exactly the same for learning different rules, so we do not implement SyGuS or any discrete searching based algorithms to learn the logic rules.

---

> > ### Author Response · Authors · 2022-11-17
> > **Author Response 2**
> >
> > If we draw an analogy between our model and neural networks, then the construction of our networks is analogous to defining the structures of NNs, while the rule generation process of our model is analogous to learning the parameters of NNs. A key difference between our model and the SyGuS methods is that when learning different rules, the structure of the network in our model stays invariant while for SyGuS methods they need to generate different structures to capture different rules. Also, the construction procedure provided in this paper is extendable, meaning that one can repeat the construction procedure for arbitrary times to obtain more expressive networks. We have also rewrite the construction steps to improve clarity.
> >
> > > the empirical evaluation shows very little improvement. Particularly, around 1% or less improvement on three knowledge graph completion datasets (i.e., Kinship, FB15k-237, WN18RR).
> >
> > We admit that the improvement of empirical evaluation in KG completion tasks are rather little, which is a drawback of our work. We argue that the main goal of us is to provide a general differentiable learning framework that is able to handle a large variety of tasks. Let's consider the 3 experiments in this paper.
> > 1. For KG completion tasks, a previous rule learning work, RNNLogic, gives competitive results against our work. However, it's unlikely to extend RNNLogic to be capable of solve the ILP and CLUTRR tasks because (1) RNNLogic+ uses a RNN to generate rules, which naturally restrict the search space into chain-like horn clauses; (2) If we somehow extend the original rule generator to be able to generate more complex rules, since they use EM algorithm to train the rule generator together with an approximation when computing the posterior, to avoid local optimal they have to use a rule miner to identify useful logic rules first to guide the training direction, so the rule miner also requires to be modified; (3) RNNLogic+ uses a rather complex predictor (rule embedding + MLP) compared with original RNNLogic, and this might be problematic when less data are available. (4) At last, RNNLogic is built to solve the queries $(h,r,?)$, and they model the probability as $p(a|(h,r,?))$ which is normalized across all possible tail entities, thus for relation prediction tasks, direct comparation across different relations is infeasible in this framework. \
> > We have tried to modify RNNLogic to make it work on the relation prediction task CLUTRR and chose the best results. The results are as follows. There are total $40$ relations in the dataset, so randomly choosing provides $0.025$ accuracy.
> >
> > |           |  4-Hop   | 5-Hop  | 6-Hop | 7-Hop | 8-Hop | 9-Hop | 10-Hop
> > |  ----     |  ----  | ----  | ------ | ------- | ------- | ------- | ------- |
> > |   RNNLogic     | 0.091  | 0.066 | 0.071 | 0.061 | 0.054 | 0.043 | 0.046
> >
> > Also, in fact, in the earlier stages when developing the algorithm, we have tried another routine, where we jointly use knowledge graph embedding and logic rules to make predictions. The results are rather good as follows, but we finally discard them finally because our overall goal is to provide a general method for learning logic rules.
> >
> > |        | MR    | MRR   | H@1    | H@3    | H@10    |
> > |  ----  | ----  | ----  | ----  | ----  | ----  |
> > | UMLS   | 1.9 | 0.920 | 0.875 | 0.961 | 0.984 |
> > | Kinship| 2.3 | 0.813 | 0.725 | 0.875 | 0.966 |
> >
> > 2. Another work that is competitive with ours in the CLUTRR tasks is R5. In the Section 4 and Appendix E of the R5 paper the authors discuss why R5 is not suitable for KG completion tasks, as "*KBC is different from the general relation prediction tasks investigated in this paper*", because R5 targets at finding the correct relation given the head and tail, i.e. $(h,?,t)$, and for KBC tasks "*since the relation $r$ is given in the query, rather than deciding the relation type between a head entity $h$ and a candidate tail $t\* $ like what R5 might do, a better choice would be directly determining whether the single relation $r$ exists between $h$ and $t\*$*", and thus as the authors of R5 says, "*It is also non-trivial to extend our research to tackle KBC tasks*". More reasons about why R5 is infeasible for KBC problems can be seen at the paragraph 3 in Appendix E of the paper [R2].

---

> > > ### Author Response · Authors · 2022-11-17
> > > **Author Response 3**
> > >
> > > 3. For ILP tasks, $\partial$ILP is also able to solve these tasks. However, as we mentioned before, $\partial$ILP uses different templates specifically designed for different tasks, and they are not scalable to larger knowledge graphs.
> > >
> > > Overall, we are keeping updating the paper to improve it's clarity and preciseness. The ideas presented in this paper surely have been explored by many recent papers, but we believe that we also take a step further: we discovery the ability to learn more complex rules as well as finding methods for explicitly learning symbolic structures via continuous optimization. The empirical improvement in KG completion tasks is not much significant, but we provide a general algorithm that is able to deal with different tasks, and in these different tasks the performance of our model is competitive (even achieve better results) against previous state-of-the-art models from different fields ($\partial$ILP for ILP tasks, R5 for CLUTRR, i.e. relation prediction tasks, RNNLogic for rule-based KG completion tasks).
> > >
> > > [R1] Luc De Raedt, Sebastijan Dumancic, Robin Manhaeve, Giuseppe Marra: From Statistical Relational to Neuro-Symbolic Artificial Intelligence. IJCAI 2020: 4943-4950.
> > >
> > > [R2] Shengyao Lu, Bang Liu, Keith G. Mills, Shangling Jui, and Di Niu. R5: Rule discovery with reinforced and recurrent relational reasoning. ICLR 2022.

---

### Official Review · Reviewer_qAry · 2022-10-30

**Confidence:** 2
**Correctness:** 3
**Technical Novelty And Significance:** 3
**Empirical Novelty And Significance:** 3
**Recommendation:** 5

**Clarity, Quality, Novelty And Reproducibility:**

Some of the technical details are unclear to me and the readability of this work needs to be further improved.


**Strength And Weaknesses:**

This work aims to bridge between differentiable programming and symbolic reasoning which has been an important research topic. However, here are a few concerns/confusions when I go through this work:

- Figure 1 is very helpful for understanding the differentiable logic program. Still, it seems that the structure of the learned is not arbitrary since it depends on how the network is defined. Can the authors further illustrate how expressive are the learned logic rules and what are the limitations?
- Also, I wonder how are the edges are chosen in the example in Figure 1. For example, why \phi_2 choose the edge connected to Red but not the edge connected to Blue, is there some threshold for for the weights to decide which edge results in the learned logic rule? It is not clear in Sec. 3 or Sec. 4 how this is achieved.
- For Prop. 4.2, it is unclear to me what these assumptions mean. Specifically, for the assumption (1), the notation is confusing that on the left hand side the x seems to be a variable while on the left hand side x seems to be taken over all possible assignments to the variables. For assumption (2), it is unclear to me what does "remove all queries Q(x) = 1" and "proved true by previous ..." mean. The authors might want to provide a formal description here to improve readability.
- In Equation (7), what are the alpha's? Why do we need alpha and how are alpha chosen? It does not seem that they are learnable parameters.
- The empirical evaluation is extensive. Still, it is not obvious that the proposed framework is more advantageous over RNNLogic+ in Table 3.

Minor issues:
- in Preliminary, the same notation \mathcal{V} is used for bot the set of constants and also the set of variables.
- at the first line of Equation (1), should it be exist y instead of exist z?
- \sigma in Equation (8) is not defined.

**Summary Of The Paper:**

This work proposes a differentiable logic programming framework that aims to learn first-order logic structures and weights. This framework further supports probabilistic reasoning by performing probabilistic forward chaining. Some theoretical guarantees on error rate and solution optimality are shown. Empirical evaluations of the proposed framework are carried out in various tasks including inductive logic programming, systematic reasoning and knowledge graph completion.

**Summary Of The Review:**

My main concern is the readability of this work.

---

> ### Author Response · Authors · 2022-11-17
> **Author Response 1**
>
> We thank the reviewer for the insightful comments! We are keeping updating the paper to improve its clarity and readability. The reviewer is welcomed to check the revised paper.
>
> > Figure 1 is very helpful for understanding the differentiable logic program. Still, it seems that the structure of the learned is not arbitrary since it depends on how the network is defined. Can the authors further illustrate how expressive are the learned logic rules and what are the limitations?
>
> The expressiveness of the learnt rules is determined by two factors: the generative definition (grammar) of FOL and the construction steps of the network. We have updated Section 4.2 of the paper to provide two standard methods for constructing the network namely LP-tree and LP-layer. For any logic classifiers $\varphi$ constructed by recursively applying its grammar for $N$ times, a LP-tree of depth $N$ (with $O(2^N)$ LPs) is able to represent $\varphi$ (where the depth of the tree is equal to $N$), while the average size needed to represent $\varphi$ is $O(N)$. (considered as a random spanning tree with depth $O(\log n)$). A LP-layer of $N$ layers (with a total number of $N$ LPs) is always able to represent any $\varphi$.
>
> The limitations comes from the generative definition (grammar). If the target logic rule we want to learn cannot be constructed by the grammar, then LPs cannot learn such rule either. Otherwise, with enough layers, we can learn such rules.
>
> We have updated Figure 1 to make it consistent with the actual structures used in the experiment. The reviewer is welcomed to check it. We have also added a proposition in Section 4.2 discussing the expressiveness of LPs.
>
>
>
>
> > Also, I wonder how are the edges are chosen in the example in Figure 1. For example, why \phi_2 choose the edge connected to Red but not the edge connected to Blue, is there some threshold for for the weights to decide which edge results in the learned logic rule? It is not clear in Sec. 3 or Sec. 4 how this is achieved.
>
> Which edges are chosen are learnt by the model. To be more specific, in all experiments we use the same network to learn different rules. In Figure 1 for judging $\mathrm{Blue}$ or $\mathrm{Red}$, there's a corresponding attention vector w.r.t. the predicates $\mathrm{Red}$ and $\mathrm{Blue}$ when evaluating $\psi$, roughly in the form of $w_1 ~ \mathrm{Red(x)}+w_2 ~ \mathrm{Blue(x)}$, where $w_1+w_2=1,w_1\geq 0,w_2\geq 0$. The proxy problem proposed in Section 4.4 guarantee the sparisity of the learnt attentions, which means when training completes, the attention vector $\mathbf{w}$ becomes one-hot and we have $w_1$=1 and $w_2=0$, thus the model choose the edge connected to $\mathrm{Red}$ rather than $\mathrm{Blue}$. In reality, choosing which edge is done by using $\mathrm{argmax}$.
>
> > For Prop. 4.2, it is unclear to me what these assumptions mean. Specifically, for the assumption (1), the notation is confusing that on the left hand side the x seems to be a variable while on the left hand side x seems to be taken over all possible assignments to the variables. For assumption (2), it is unclear to me what does "remove all queries Q(x) = 1" and "proved true by previous ..." mean. The authors might want to provide a formal description here to improve readability.
>
> We have rewrite Prop. 4.2 to make it more clear. The reviewer is welcomed to check it again. The goal of Prop. 4.2 is to demonstrate the general learning procedure of our model. Just like boosting methods, we learn logic rules recursively: we first learn a logic rule using Eq. (4), and then we decrease the weights of correctly predicted data instances while increasing those of wrongly predicted ones, and do this again until enough rules are learnt.
>
>
>
> > In Equation (7), what are the alpha's? Why do we need alpha and how are alpha chosen? It does not seem that they are learnable parameters.
>
> $\alpha\in(0,+\infty)$ is the hyperparameter used to help keep the sparisity of the model. In the experiments of this paper, $\alpha$ is set to $1$ or $2$, decided via the validation data. In fact, as shown in the ablation studies, by simply setting $\alpha=1$ the model also works well.

---

> > ### Author Response · Authors · 2022-11-17
> > **Author Response 2**
> >
> > > The empirical evaluation is extensive. Still, it is not obvious that the proposed framework is more advantageous over RNNLogic+ in Table 3.
> >
> > We admit that the performance of our model is close to RNNLogic+ on WN18RR and FB15k-237 datasets. One thing we want to clarify is that we are trying to present a general rule learning model that can handle a variety of tasks. RNNLogic+ is a great work, but it is not suitable for CLUTRR and ILP tasks in this paper. It is not easy to extend this work to be able to do this because (1) RNNLogic+ uses a RNN to generate rules, which naturally restrict the search space into chain-like horn clauses; (2) Even if we somehow extend the original rule generator to be able to generate more complex rules, since they use EM algorithm to train the rule generator together with an approximation when computing the posterior, to avoid local optimal they have to use a rule miner to identify useful logic rules first to guide the training direction, so the rule miner also requires to be modified; (3) RNNLogic+ uses a rather complex predictor (rule embedding + MLP), compared with original RNNLogic, and this might be problematic when less data are available. (4) At last, RNNLogic is built to solve the queries $(h,r,?)$, and they model the probability as $p(a|(h,r,?))$ which is normalized across all possible tail entities, thus for relation prediction tasks, direct comparation across different relations is infeasible in this framework.
> >
> > We have tried to modify RNNLogic to make it work on the relation prediction task CLUTRR and selected the best results. The results are as follows. There are total $40$ relations in the dataset, so randomly choosing provides $0.025$ accuracy.
> >
> > |           |  4-Hop   | 5-Hop  | 6-Hop | 7-Hop | 8-Hop | 9-Hop | 10-Hop
> > |  ----     |  ----  | ----  | ------ | ------- | ------- | ------- | ------- |
> > |   RNNLogic     | 0.091  | 0.066 | 0.071 | 0.061 | 0.054 | 0.043 | 0.046
> >
> > > in Preliminary, the same notation $\mathcal{V}$ is used for both the set of constants and also the set of variables.
> >
> > We have fixed this issue in the updated version of paper.
> >
> > > at the first line of Equation (1), should it be exist y instead of exist z?
> >
> > It is indeed $\exists z$. Equation (1) aims at providing an example of how we construct FOL formulas, and $\exists z: \varphi(x,z)\wedge\varphi(z,y)$ is a typical logic pattern, for example, $\mathrm{Grandfather}(x,y):=\exists z: \mathrm{Father}(x,z)\wedge\mathrm{Father}(z,y)$ is the common definition for $\mathrm{Grandfather}$.
> >
> > Moreover, we have updated Equation (1) to make it consistent with the actual grammar used across the experiments.
> >
> > > \sigma in Equation (8) is not defined.
> >
> > $\sigma$ is the sigmoid function, use to guarantee the results fall in $(0,1)$. We have corrected this issue in the paper.

---

### Official Review · Reviewer_qPBm · 2022-11-04

**Confidence:** 3
**Correctness:** 2
**Technical Novelty And Significance:** 2
**Empirical Novelty And Significance:** 3
**Recommendation:** 3

**Clarity, Quality, Novelty And Reproducibility:**

As described above, the work is extremely unclear and I don't think I can reproduce it. It's difficult to pin down novelty without some clarity on what has been done; certainly differentiable approximations to logical operators have been proposed previously, so at that level of detail, the work is not novel.

**Strength And Weaknesses:**

The main strength is that the proposed method seems to do well on CLUTRR and relatively well on the knowledge graph completion tasks, compared to the handful of chosen baselines for each task.

There are some significant weaknesses, however. The main weakness is that the presentation of the method is essentially unreadable and disturbingly vague at places.

To begin with, the problem statement is unclear: the input data is described as both a set of FOL statements and as a set of triples. It is not clear to me if the FOL statements are supposed to be ground atoms (as they are in the example) or contain free variables as the notation suggests, or if they can be arbitrary FOL formulas. The tasks "Recognition of Logical Patterns" and "Probabilistic Reasoning" are only described in rather vague terms: what does it mean to "explain how the answers are deduced?" What does it mean to "make decisions [some expression] based on the groundings of classifiers"?

In Proposition 4.2, what is the "precision on B"? In the formulas there, for the sums indexed by x, what does x range over? What does it mean to "remove all queries Q(x) = 1 and also proved true"? What is the "prediction model" being constructed -- what does this notation with p_w mean? Apparently after learning such models, we "remove or reduce the weights of data instances where Q(x) = phi_i(x) = 1" -- what are these weights and which is it, remove or reduce them, and by how much?

In Equations 6 & 7, it seems like some strange vector notation is being used, but the dimensions don't seem to work and I'm not certain I know what is meant by raising the vector to the power alpha.

In the section on inference, I still don't know what x ranges over; are we looking at all possible groundings? The function "sigma" is not defined. Q_0 is also not defined.

The description of constructing the networks is vague. Equation 7 gives a correspondence between connectives and the differentiable approximations; how do we obtain "output LPs" from these definitions? What does it mean for "another LP" to be "needed"? We repeat the procedure "a couple of times" -- meaning two? or just some arbitrary constant number?

In the learning section, Proposition 4.3, how do you decide when a node should be fixed? What does it mean for the solution to be "near-optimal"?  Needless to say, I don't see why the vague construction procedure satisfies these constraints, in particular the constraint of having a tree topology on the unfixed nodes. Note that the network in Figure 1 definitely does not have a tree topology. Actually, this raises a concern, that to satisfy this requirement, the method might need to explicitly represent the tree of possible logical expressions. Needless to say, this would not seem to offer any computational advantage over traditional ILP methods.

Moving on, in the experiments, there are a few fishy features. For one, the authors chose to write their numbers in boldface when there are other methods that seem to have essentially equivalent performance as far as I can tell. For some reason the authors give their method's numbers to three decimal places when the stated accuracy clearly only warrants two. Also, for some reason the knowledge graph completion tasks are only examined on subsets of the baselines for unknown reasons.

Ultimately, my main concern with the experiments is that the vagueness of the description of the method may be covering some kind of handcrafting of solutions for the various benchmark problems. At several points, e.g., the definition of the "update" function, the choice of structure, etc., the text suggests that there are many possible choices. If these choices are made by the experimenter individually per benchmark, it calls the validity of the results into question.

**Summary Of The Paper:**

The paper proposes a method for an ILP-style rule-learning problem by using gradient descent over differentiable approximations to the various connectives in the language under consideration. The method is empirically tested on a suite of ILP tasks, the CLUTRR benchmark, and some knowledge graph completion tasks.

**Summary Of The Review:**

The presentation of the proposed method is too vague to be useful. At times, the presentation seems to deliberately avoid committing to details of the method. This also raises questions about the validity of the experiments, specifically whether or not a single, common algorithm is being evaluated.

---

> ### Author Response · Authors · 2022-11-17
> **Author Response 1**
>
> We thank the reviewer for the insightful comments!
>
> We admit that the current writing of this paper is quite vague and unclear. One thing we want to clarify is that we are not deliberately avoiding committing to details of our method. However, we admit that we did tried to make this work looks more general and flexible. We have updated the draft to improve the clarity and readability, as well as addressing your concerns. A list of updates is available in the official comment. Another thing we want to clarify first is that in all the experiments of this paper, no handcrafting solutions are used. We now discuss about all you questions.
>
> ## Paragraph 1, about problem statement
> > the input data is described as both a set of FOL statements and as a set of triples. It is not clear to me if the FOL statements are supposed to be ground atoms (as they are in the example) or contain free variables as the notation suggests, or if they can be arbitrary FOL formulas.
>
> We have updated the paper to make the description of the input data consistent with the standard ILP problems across the sections. All the input data are now described as a set of instances $(\mathcal{S_B},\mathcal{S_P},\mathcal{S_N})$ where $\mathcal{S_B}$ is the set of background atoms, $\mathcal{S_P}$ the set of positive atoms and $\mathcal{S_N}$ the set of negative atoms. All the FOL statements in the data are ground atoms. To be more specific, in the experiments of this paper, we have 3 different kinds of input data formats, including (1) ILP problems, (2) CLUTRR and (3) knowledge graphs.
>
> 1. A typical ILP problem can be described as a tuple $(\mathcal{S_B},\mathcal{S_P},\mathcal{S_N})$ (see paper R1) naturally. To be more specific, consider the even numbers recognition tasks, where we have $\mathcal{S_B}=\{zero(0),succ(0,1),succ(1,2),...\}$, $\mathcal{S_P}=\{even(0),even(2),...\}$ and $\mathcal{S_N}=\{even(1),even(3),..\}$.
> 2. The CLUTRR dataset is composed of a number of examples, each composed of background family relationships $\mathcal{S_B}$, and a target such as $\mathrm{Brother(Tom,Bob)}$. We use negative sampling to generate negative atoms $\mathcal{S_N}$. Then, each such example  corresponds to an instance, e.g. $(\mathcal{S_B},\{\mathrm{Brother(Tom,Bob)}\},\mathcal{S_N})$.
> 3. A knowledge graph $\mathcal{G}$ can be seen as a collection of facts. For each fact $r(a,b)$, i.e. an edge from $\mathcal{G}$, we build a positive instance $\mathcal{S_P}=\{r(a,b)\}$. We sample negative atoms by randomly breaking the head / tail entities of positive atoms, such as $\mathcal{S_N}=\{r(a,e_1),r(e_2,b)\}$. The graph $\mathcal{G}$ is naturally regarded as background assumptions $\mathcal{S_B}$.
>
> > The tasks "Recognition of Logical Patterns" and "Probabilistic Reasoning" are only described in rather vague terms: what does it mean to "explain how the answers are deduced?" What does it mean to "make decisions [some expression] based on the groundings of classifiers"?
>
> We have revised the paper to describe these concepts more clearly. To be more specific, the task "Recognition of Logical Patterns" is now described as "Rule Mining". Consider an example where we have background assumptions $\{\mathrm{Father}(a,b),\mathrm{Father}(b,c)\}$. Then, the logic rule $\mathrm{GrandFather}(x,y):=\exists z:\mathrm{Father}(x,z)\wedge\mathrm{Father}(z,y)$ explains how the answer "$\mathrm{GrandFather}(a,c)$" is deduced, and this answer is obtained by grounding the rule body with $x\leftarrow a$, $z\leftarrow b$ and $y\leftarrow c$.
>
> We have updated the paper to make the expressions more clear. The tasks are now described as "Rule Mining" and "Probabilistic Reasoning".
>
> ## Paragraph 2, About Proposition 4.2
> Proposition 4.2 has a direct connection with Example 4.1. The goal of the Proposition is to loosely justify our whole rule-learning framework, which is somewhat like boosting methods: we first learn a rule on the original data, and then reweight the data and learn rules again. We'll discuss the details in the following.
>
> > what is the "precision on B"? In the formulas there, for the sums indexed by x, what does x range over?
>
> $\mathbf{x}$ ranges over all possible values that it could take, which means that we are looking for all possible groundings. Let $\mathcal{V}$ be the set of all constants and $\mathbf{x}$ be a $d$-ary tuple, then sums mean $\sum_{\mathbf{x}\in\mathcal{V}^d}$. For example, if the input data is a knowledge graph with nodes set $\mathcal{V}$, then all queries $Q$ are binary predicates corresponding to edges in the graph, and $\mathbf{x}$ ranges over $\mathcal{V}^2$, which is the set of all possible node tuples.
>
> To discuss the precision on B, suppose a rule $Q(\mathbf{x})\leftarrow \varphi(\mathbf{x})$. The precision of the rule is
>
> $$\frac{number ~ of ~ \mathbf{x}:Q(\mathbf{x})~and ~\varphi(\mathbf{x}) ~ are ~ both ~ true}{number ~ of ~ \mathbf{x}:\varphi(\mathbf{x}) ~ is ~ true}$$
>
> We have updated the paper to address these problems more clearly.

---

> > ### Author Response · Authors · 2022-11-17
> > **Author Response 2**
> >
> > > What does it mean to "remove all queries Q(x) = 1 and also proved true"? What is the "prediction model" being constructed -- what does this notation with p_w mean? Apparently after learning such models, we "remove or reduce the weights of data instances where Q(x) = phi_i(x) = 1" -- what are these weights and which is it, remove or reduce them, and by how much?
> >
> > Generally the rule learning procedure is analogous to boosting methods. Let $(\mathcal{S_B},\mathcal{S_P},\mathcal{S_N})$ be the input data. Initially we start with an empty rule set $\Phi=\emptyset$, and we assign each positive / negative instance $Q(\mathbf{x})\in\mathcal{S_P}$ and $Q(\mathbf{x})\in\mathcal{S_N}$ with a weight
> > $w_Q(\mathbf{x})=1$. Then, at each iteration, we first learn a rule $Q\leftarrow\varphi$ defined in Eq. (4), then we apply the rule on $\mathcal{S_B}$, and (1) if $\varphi(\mathbf{x})=1$ and $Q(\mathbf{x})\in S_P$, we set $w_Q(\mathbf{x})\leftarrow\tau_1 w_Q(\mathbf{x})$; (2) if $\varphi(\mathbf{x})=1$ and $Q(\mathbf{x})\in S_N$, we set $w_Q(\mathbf{x})\leftarrow \tau_2 w_Q(\mathbf{x})$. We add $\varphi$ to the rule set $\Phi$ and repeat this procedure again. Here, $\tau_1$, $\tau_2$ are hyperparameters, and is selected in $(\tau_1,\tau_2)=\{(0,1),(\frac{1}{5},5),(1,1)\}$. We repeat this procedure for $400$ times on KG completion tasks, and for other tasks we repeat until all instances in $\mathcal{S_P}$ are proved.
> >
> > The reviewer is also welcomed to check Sec. 4.1 of the paper where we have rewrite the  proposition and the rule learning procedure.
> >
> > ## Paragraph 3, About Equations 6 & 7
> > > it seems like some strange vector notation is being used, but the dimensions don't seem to work.
> >
> > Recall that the $\varphi(\mathbf{x})$ and $\psi(\mathbf{x})$ are scalars. $\mathcal{F}_1(\mathbf{x})$, $\mathcal{F}_2(\mathbf{x})$, ... are also scalars. Thus, $[\mathcal{F}_1(\mathbf{x}),\mathcal{F}_2(\mathbf{x}),...]$ is a row vector -- let's say it's dimension is $L$ for now. Then, $\theta$ is also a column vector of dimension $L$, and $\theta^\alpha$ is an element-wise power of $\theta$. $\theta^T$ stands for the transpose of $\theta$.
> >
> > We have rewrite Equations 6 & 7 to make these statements more clear.
> >
> > > I'm not certain I know what is meant by raising the vector to the power alpha.
> >
> > One key difference between our model and other differentiable rule-learning model is that after training, our model explicitly finds logic rules. $\alpha$ is used to guarantee the sparsity of the learnt attention vectors $\mathbb{\theta}$ and $\mathbb{w}$ so that we can explicitly find the structures of logic rules. Moreover, as shown in the experiments, by setting $\alpha=1$ the model also works well. (as shown in Table 7).
> >
> > ## Paragraph 4, About inference
> > > I still don't know what x ranges over; are we looking at all possible groundings?
> >
> > Yes. $\mathbf{x}$ ranges over all possible groundings. We have revised the paper to address this issue.
> >
> > > The function "sigma" is not defined.
> >
> > $\sigma$ is the sigmoid function, used to guarantee that the results fall in $(0,1)$. We have changed "$\sigma$" into "$\mathrm{sigmoid}$" in the paper to clarify this.
> >
> > > Q_0 is also not defined.
> >
> > In fact, this is because of our mistakes: we improperly used $P$ and $Q$ interchangeably, where $Q\in\mathcal{P}$ is inclined to represent "query predicates" and $P\in\mathcal{P}$ is inclined to represent "base predicates". So it's okay to think that $Q_0$ is initialized the same as $P_0$, which is the same as observed atoms. We have updated the inference section to correct this problem.
> >
> > ## Paragraph 5, About constructing the networks.
> > We have completely rewrite this section. Now, the methods for constructing the network is in section 4.2. Hope the new version would address these problems.
> >
> > Moreover, we have described two versions of construction procedures more formally. The first one, namely LP-tree, is the original one used in the experiments, which have a tree structure. The second one, namely LP-layer, is an additional one proposed to address the concerns about the size of the network. We have discussed the expressiveness of both structures in Sec. 4.2, as well as adding additional experiments to show the effectiveness of LP-layer.
> >
> > We have added additional experiments of LP-layer on ILP and CLUTRR datasets, and it provides similar results with LP-tree. We have also studied the sparsity of the learnt model, where we rerun LP-layer on the three experiments with $\alpha=1$ and check the learnt parameters to determine whether the model exactly captures a logic classifier. The results are shown in Appendix G.

---

> > > ### Author Response · Authors · 2022-11-17
> > > **Author Response 3**
> > >
> > > > how do we obtain "output LPs" from these definitions? What does it mean for "another LP" to be "needed"?
> > >
> > > In the revised version of Sec. 4.2, we have defined how we generate the network from scratch. Let's discuss the construction procedure more clearly.
> > >
> > > For the tree-structured LPs, we first create a LP $\psi$ as the root of the tree, and $\psi$ is the "output LP". To generate the tree structure, let's first look at the evaluation procedure of $\psi$. In Eq. (7), there is a "$\psi'$" emerged in the definition of $F_i$, but we haven't assigned $\psi'$ with an actual LP yet in the network. We refer to $\psi'$ here as "another LP that is needed". We create a new LP $\psi_{new}$ to replace $\psi'$ in the evaluation procedure of $\psi$ and refer to $\psi_{new}$ as $\psi$'s child.
> > >
> > > The reviewer is also welcomed to check the revised construction procedure in Sec. 4.2. Hope this revision would address these problems.
> > >
> > > > We repeat the procedure "a couple of times" -- meaning two? or just some arbitrary constant number?
> > >
> > > In the experiments, it is $5$ for ILP problems, $3$ for CLUTRR and KG completion. We have addressed this in the newly updated Sec. 5. But these numbers are not necessary: we have added an additional experiment for ILP problems and CLUTRR where we simply repeat the construction procedure for more times, including $\{3,5,10\}$, and the results are shown in Appendix G. With $3$ layers some of the ILP problems cannot be solved because they need a longer reasoning depth. Also, we have performed the ablation study on KG where we only reserve rules of length $2$, $3$ and $4$ respectively. The results are shown in Table 4 of the paper.
> > >
> > > ## Paragraph 6, About learning.
> > > > how do you decide when a node should be fixed?
> > >
> > > We have rewrite the proxy problem to stay consistent with the updated construction steps as well as avoiding these annoying concepts.
> > >
> > > > What does it mean for the solution to be "near-optimal"?
> > >
> > > We do not say the solution is global optimal because (1) the LPs can only capture logic classifiers within their expressive power and (2) the evaluation of logic classifiers $\varphi$ is slightly different from LPs $\psi$: we always have $\varphi(\mathbf{x})\in\{0,1\}$, but we have $\psi\in[0,+\infty)$. For example, if we have $\varphi(x):=\exists y:\mathrm{Neighbor}(x,y)$, then the corresponding $\psi(x)$ is equal to the amount of neighbors that $x$ have. So even when $\psi$ explicitly captures $\varphi$, the evaluation of $\psi$ is different from $\varphi$, and is more like an instance of relaxed Lukasiewicz t-conorm.
> > >
> > > Besides, we have added an additional experiment where we restrict the range of $\psi$ to be $[0,1]$, as $\mathrm{min}\{\psi(\mathbf{x}),1\}$. Then, minimization of the proxy problem is exactly the global optimal solution. The results of comparation experiment on KG are shown in Appendix G. On ILP and CLUTRR, setting the restriction or not doesn't have a significant influence on the results.
> > >
> > > > I don't see why the vague construction procedure satisfies these constraints, in particular the constraint of having a tree topology on the unfixed nodes.
> > >
> > > The construction procedure is corresponded to generating a tree from its root to the leaf nodes. We start with only one root node in the tree, and then we recursively add new nodes to be the children of the leaf nodes.
> > >
> > > We have rewrite the construction steps of tree-structured LPs to state this more clear in Sec. 4.2.
> > >
> > > > The network in Figure 1 definitely does not have a tree topology.
> > >
> > > Actually the network in Figure 1 does not have a tree topology. The target of Figure 1 is to intuitively illustrate the basic ideas of this work, so it does not strictly obeys with the construction procedure described in Sec. 4.
> > >
> > > We have updated the Figure 1. Now it stays the same with the construction procedure provided in Sec. 4 and the generative definition (now referred to as grammar) provided in Eq. (1) of Sec. 3, as well as having a tree topology.
> > >
> > > > Actually, this raises a concern, that to satisfy this requirement, the method might need to explicitly represent the tree of possible logical expressions.
> > >
> > > There are two factors that influence how the tree grows. To make it more clear, assume the grammar is $\varphi(x):=P(x) ~ | ~ \varphi(x,x) ~ | ~ \exists y:\varphi(y)\wedge\varphi(x,y)$. Then, to recursively construct a logic classifier $\varphi_0$ from this grammar, there are two major problems to consider, which are:
> > > 1. Choice of formats: which format should $\varphi_0$ take? Should $\varphi_0$ be some base predicate $P$, or another binary classifier $\varphi_1(x,x)$, or be in the form of $\exists y:\varphi_2(y)\wedge\varphi_3(x,y)$? Suppose there are total $K$ choices.
> > > 2. Choice of predicates: When $\varphi_i$ is some base predicate, which predicate should it exactly choose? Clearly there are $|\mathcal{P}|$ different choices.

---

> > > > ### Author Response · Authors · 2022-11-17
> > > > **Author response 4**
> > > >
> > > > The tree structures need to represent the choice of formats, but they don't need to represent the choice of predicates. Suppose $\varphi_0$ is constructed by recursively applying the grammar for $N$ times, in worst cases the size of the tree is $O(|\mathcal{P}|N+K^N)$, i.e., it grows linearly with $|\mathcal{P}|$ but exponential for representing $K$ different choices.
> > > >
> > > > Besides, to address this concern, we have provided another construction procedure that generated layer-structured LPs (namely LP-layer). This no longer satisfies the tree constraints, but we have also added additional experiments to show that the layer-structured LPs also converges well in Appendix G.1. The advantage is that the size of LP-layer always grows linearly w.r.t. $N$. In fact, suppose $\varphi_0$ is constructed by recursively applying its grammar for $N$ times, LP-layer with $N$ nodes is able to capture $\varphi_0$. We have added discussion about LP-layer and their expressiveness in Sec. 4.2.
> > > >
> > > > > this would not seem to offer any computational advantage over traditional ILP methods.
> > > >
> > > > We believe that there are certainly many more efficient rule-learning algorithms compared to our method, but our method is also scalable. To show our computational complexity theoretically, learning one rule of length $L$ on knowledge graphs takes $O(SL|V|N^{(L)}N^{(1)})$, where $S$ is the number of optimization steps, which is less than $15$ in our experiments; $|V|$ is the number of all nodes; $N^{(k)}$ is the average number of nodes in $k$-hop neighborhood subgraphs. To show our computational complexity empirically, we run several experiments with our approach and RNNLogic on the same machine with the same -O3 compile options and compare the time used. For RNNLogic, the hyperparameters we use are the same as provided in their official codes.
> > > >
> > > > |           |  UMLS   | Kinship  |
> > > > |  ----     |  ----  | ----  |
> > > > |   DLP     | 34.73s  | 53.89s |
> > > > |  RNNLogic | 317.64s  | 362.92s |
> > > > | RNNLogic-torch | 5263.02s | 4269.48s |
> > > >
> > > >
> > > > ## Paragraph 7, About experiments.
> > > > > For one, the authors chose to write their numbers in boldface when there are other methods that seem to have essentially equivalent performance as far as I can tell.
> > > >
> > > > We have updated the paper to correct these issues.
> > > >
> > > > > For some reason the authors give their method's numbers to three decimal places when the stated accuracy clearly only warrants two.
> > > >
> > > > We have rerun the experiments for 10 times and recalculated the results.
> > > >
> > > > > Also, for some reason the knowledge graph completion tasks are only examined on subsets of the baselines for unknown reasons.
> > > >
> > > > On knowledge graph completion we compare our approach with other rule-based approaches. Generally, embedding methods achieve better results than rule-based methods, but they provide poor explainability, so it is unfair to compare with these methods. For example, to the best of our knowledge, **NO rule-based methods** is able to beat NBFNet [R2] in KG completion tasks, as it gets $0.415$ MRR on FB15k-237 and $0.551$ MRR on WN18RR. Also, many of the baselines in this paper (such as R5,RNN,LSTM) are not suitable for solving KG completion tasks.
> > > >
> > > > ## Paragraph 8, About handcrafting solutions
> > > > > Ultimately, my main concern with the experiments is that the vagueness of the description of the method may be covering some kind of handcrafting of solutions for the various benchmark problems.
> > > >
> > > > In all the experiments, no handcraft solutions or unnecessary restrictions of search spaces of rules are provided to cheat the experiments. We have updated the paper so that the experiment details, including the structure of networks we used, are discussed.
> > > >
> > > > Across all the experiments, the grammar (generative definition) we use are all the same: $\varphi(x):=A(x) ~ | ~ \varphi(x,x) ~ | ~ \exists y:\varphi(y)\wedge\varphi(x,y)$, and $\varphi(x,y):=A(x,y) ~ | ~ \varphi(x)\wedge\varphi(y)\wedge\varphi(x,y) ~ | ~ \exists z:\varphi(x,z)\wedge\varphi(z)\wedge\varphi(z,y)$. When data contains no unary predicates, we simply apply an invented predicate $P_{invent}(x)\equiv 1$. This leads to the fact that for KG completion tasks, the grammar is essentially corresponded to chain-like rules due to the absent of unary predicates.
> > > >
> > > > Besides, the network construction steps are the same across the experiments, and they are discussed in Sec. 4.2 of the updated paper. In the experiments when learning different rules, the structure of the network stays the same.
> > > >
> > > > > At several points, e.g., the definition of the "update" function, ., the text suggests that there are many possible choices ... it calls the validity of the results into question.
> > > >
> > > > For all experiments except WN18RR, the update function is a weighted sum, i.e., $\mathrm{Update}(\mathbf{x})=\sum_i w_i\psi_i(\mathbf{x})$. For WN18RR, the update function is a MLP as follows: $N\rightarrow 12\rightarrow 6\rightarrow 4\rightarrow 1$, but this is not necessary and others also do well. We use relu as the activation function of MLP.

---

> > > > > ### Author Response · Authors · 2022-11-17
> > > > > **Author response 5**
> > > > >
> > > > > Besides, we have added an additional experiment on WN18RR where we also set the update function to be the weighted sum. The result is as follows.
> > > > > | MR      | MRR | H@1 | H@3 | H@10
> > > > > | ----------- | ----------- | ----------- | ----------- | ----------- |
> > > > > | 7217      | 0.477       | 0.434 | 0.501 | 0.559
> > > > >
> > > > > The results are now lower than RNNLogic+ but higher that RNNLogic. This is because the only difference between RNNLogic+ and RNNLogic is that RNNLogic uses a weighted sum as predictor, while the predictor of RNNLogic+ is more complex, including a rule embedding vector for each learnt rule and a MLP to aggregate the results.
> > > > >
> > > > > As for the choice of structure, we have discussed about it both in the revised paper and the above discussions, which are simple and consistent across the experiments. To clarify, the hyperparameters, model structure, other model configurations, etc. are exactly the same for solving all unary ILP problems; configurations for solving all binary ILP problems are the same; configurations for solving all CLUTRR datasets are the same. For KG completion, the hyperparameters are chosen via the validation data.
> > > > >
> > > > > ## About novelty.
> > > > > We have again checked the related works, also carefully read the survey provided by reviewer kdMZ. We now discuss the position that our work should be in the rule-learning fields.
> > > > >
> > > > > From the first glance our approach is very similar to other differentiable approaches such as NeuralLP, DRUM and so on. In fact, in the recent years, many differentiable rule learning methods have been proposed. We have also carefully checked the papers that cited NeuralLP, and as far as we know, these differentiable learning methods have two major problems:
> > > > >
> > > > > (1) They are mainly concentrate on the KG completion tasks, which means that although they have good scalability, the types of rules they can learn are largely restricted where predicates are binary.
> > > > >
> > > > > (2) Another problem is mentioned both in the Related Work section of this paper and in the Section 6 of the survey [R5]. They do not learn logic rules explicitly. The basic ideas of NeuralLP, DRUM and their extensions is to approximate the evaluation of all possible chain-like horn rules by exchanging the order of multiplication and summation. By doing so, the model is differentiable, but the expressiveness is largely hurt: it's hard to tell what their parameters actually means or what rule structures they have found because the learnt parameters are in fact dense vectors. Compared to these methods, our method is able to learn sparse solutions by optimizing on the continuous proxy problem where each local optimal of the continuous proxy problems explicitly reveals the symbolic structure of a logic rule. The experiments also show the effectiveness of our sparse solutions, compared with the dense vector representations of NeuralLP, DRUM, etc.
> > > > >
> > > > > To summary, compared with these methods, our method (1) is able to learn rules of arbitrary structures with arbitrary arity; (2) is able to find sparse solutions where explicit symbolic structures are learnt.
> > > > >
> > > > >
> > > > >
> > > > >
> > > > > [R1] Richard Evans and Edward Grefenstette. Learning explanatory rules from noisy data. Journal of Artificial Intelligence Research, 61:1–64, 2018.12
> > > > >
> > > > > [R2] Zhu, Zhaocheng et al. “Neural Bellman-Ford Networks: A General Graph Neural Network Framework for Link Prediction.” NeurIPS (2021).
> > > > >
> > > > > [R3] Wang, Po-Wei et al. “Differentiable learning of numerical rules in knowledge graphs.” ICLR (2020).
> > > > >
> > > > > [R4] Yang, Yuan and Le Song. “Learn to Explain Efficiently via Neural Logic Inductive Learning.” ICLR (2020).
> > > > >
> > > > > [R5] Luc De Raedt, Sebastijan Dumancic, Robin Manhaeve, Giuseppe Marra: From Statistical Relational to Neuro-Symbolic Artificial Intelligence. IJCAI 2020: 4943-4950.
> > > > >
> > > > > [R6] Sadeghian, Ali et al. “DRUM: End-To-End Differentiable Rule Mining On Knowledge Graphs.” NeurIPS (2019).

---

> ### Author Response · Authors · 2022-12-09
> **Thank you for your suggestions!**
>
> Dear reviewer qPBm,
>
> Thank you very much for your helpful feedback and suggestions, they helped us to improve the manuscript even if it is not finally accepted by the conference. We tried to carefully address all of your concerns in our response and the updated manuscript. Please let us know if you have any further questions, and we are very happy to follow up!
>
> Thank you for your time!

---

### Author Response · Authors · 2022-11-17
**List of Updates**

We thank the reviewers for their valuable suggestions. Overall, we summary the common problems as follows:
- Clarity and readability.
- Details about the the model construction.
- Empirical evaluation.

We have revised the draft to address these problems. A detailed list of updates is shown as follows.

# List of updates
Details of paper updates are as follows.

# Preliminary
- We have modified the example FOL grammar in Sec. 3 so that it is exactly what we used across all experiments in this paper.
- We have updated the problem statement to be the same as the standard definition of ILP problems in [R1].
- We have rewrite the two key problems, i.e. Recognition of Logical Patterns and Probabilistic Reasoning, to make it more clear.
- We have updated Figure 1 to address the questions of the reviewers.

### Minor updates
- We have rewrite the paragraph "Logic classifiers and logic rules" to make it more simple and clear.
- We changed the term "generative definition" into "grammar".

# Model
## Overview
- We have rewrite Proposition 4.2 to make it more clear.
- We have described the general rule learning procedure more precisely which exactly captures what we did in the experiments.

## Logic Perceptron & Inference Network
- We have rewrite the definitions of logic perceptron, inference network and the paragraph "construction of networks" together. The formulations now look different from the original ones for the sake of clarity and understandability.
- We have rewrite the section "Logic Perceptron" and "Inference Network". Now the presentations of definition of LPs and the construction methods are both described in the section "Logic Perceptron".
- We have made the construction procedure more clear.

## Learning
- We have rewrite the proxy problem, particularly the constraints. These are consistent with the updated versions of LPs.
- We have added another paragraph briefly discussing the affects of the constraints, and the consequences of violating them in the appendix.
- As discussed by the reviewers, there is a concern that to satisfy the tree constraint of the requirement, the computational efficiency of this method might be influences. We added a discussion about this problem.
- Also, we have provided another construction procedure that slightly violates the constraints of the proxy problem. Compared to the original construction procedure, this one organizes the LPs as **layers** (just like MLP) so that it is much more easy to extend: we just need to add more layers; also it could potentially save computational efficiency due to its linear structure rather than tree structure in the original construction step. The effectiveness of this construction step is shown in an additional experiment. As shown in the experiment, the layer structures
also converge well.
### Experiments
- We have fixed the issues of experiments, including boldface and precision of deviations.
- Added an additional experiment as above.

[R1] Richard Evans and Edward Grefenstette. Learning explanatory rules from noisy data. Journal of Artificial Intelligence Research, 61:1–64, 2018.12

---

### Decision · Program_Chairs · 2023-01-20

**Decision:**

Reject

**Justification For Why Not Higher Score:**

Most reviewers recommended rejecting the paper at this point. The changes made by the authors during the discussion period are are extensive enough to require a fresh round of reviewing.

**Justification For Why Not Lower Score:**

N/A

**Metareview: Summary, Strengths And Weaknesses:**

The paper proposes a differentiable approach to inductive logic programming. The reviewers have identified a number of shortcomings of the paper, including issues with clarity, limited novelty and unconvincing empirical evaluation. The authors acknowledge the issues and have made extensive changes to the paper during the discussion phase. The changes, however, are extensive enough that they would require a fresh review and reevaluation of the paper, which is outside the scope of the author feedback and discussion period and would require the paper to be resubmitted.